# Computational design and cheminformatics profiling of omeprazole derivatives for enhanced proton pump inhibition of potassium-transporting ATPase alpha chain 1

**Mahmudul Hasan**[1,2], **Md. Ifteker Hossain**[1,2], **Noimul Hasan Siddiquee**[1], **Ezaz Ahmed**[1,2], **Md Walid Hossain Talukder**[2,3], **Md Rahamatolla**[2,4], **Tasrin Nahar**[2,5], **Popy Rani Paul**[2,6], **Mahmudul Hassan Suhag**[7], **Monir Uzzaman**[2,6]*

1 Faculty of Science, Department of Microbiology, Noakhali Science and Technology University, Noakhali, Bangladesh, 2 Drug Design Division, Computer in Chemistry and Medicine Laboratory, Dhaka, Bangladesh, 3 Faculty of Science, Department of Applied Chemistry and Chemical Engineering, University of Chittagong, Chittagong, Bangladesh, 4 Faculty of Science, Department of Chemistry, University of Rajshahi, Rajshahi, Bangladesh, 5 Department of Chemistry, School of Physical Science, Shahjalal University of Science and Technology, Sylhet, Bangladesh, 6 Faculty of Science, Department of Chemistry, University of Chittagong, Chittagong, Bangladesh, 7 Faculty of Science and Engineering, Department of Chemistry, University of Barishal, Barishal, Bangladesh

* monircu92@gmail.com

## Abstract

Proton pump inhibitors are essential for treating moderate-to-severe gastroesophageal reflux, peptic ulcers, esophagitis, and related conditions by increasing gastric pH and inhibiting hydrogen ion discharge into the stomach. However, prolonged use may lead to adverse effects along with reduced efficacy. Our research investigates the strategic modification of omeprazole (OMP) derivatives to improve their binding affinity to targeted proteins, thereby enhancing their chemical reactivity, stability, and toxicity profiles. A total of 22 novel OMP analogues were designed through structural alterations, focusing on the benzimidazole and pyridine rings. The geometrical attributes of the analogues were further confirmed through spectral and quantum computational analysis based on density functional theory (DFT) and a B3LYP/6-31G+ G (d, p) basis set. The molecular docking with PTAAC1 presented that most of the analogues had similar or higher binding affinities and nonbonding interactions, including OMP3, OMP19, and OMP21, with binding energies of -7.3, -8.3, and -8.1 kcal/mol compared to the OMP at -7.1 kcal/mol. Pharmacokinetic, biological, and toxicological profiles via ADMET and PASS predictions also demonstrated increased safety and therapeutic potential. MD simulation also showed good stability of OMP3, OMP19, and OMP21 in binding to PTAAC1, and the RMSD, RMSF, ligand RMSD, rGyr, SASA, MolSA, PolSA, and hydrogen bond analysis also suggested superior drug potential compared to OMP. Additionally, the post-simulation MM/GBSA analysis revealed that OMP3 (-36.91 kcal/mol) outperformed OMP19 (-26.45) and OMP21

**Data availability statement:** All relevant data are within the manuscript and its Supporting Information files.

**Funding:** The author(s) received no specific funding for this work.

**Competing interests:** The authors have declared that no competing interests exist.

(-12.61). The protein binding site's high stability and elevated negative binding free energy value further indicate a robust compound-protein interaction with OMP3. However, principal component analysis (PCA) showed the highest variance for OMP21, accounting for 50.66%, 21.58%, and 6.51%, respectively, for PC1, PC2, and PC3. These findings could lead to the development of OMP3 and OMP21 as potential next-generation PPIs with enhanced pharmacological activity and improved side-effect profiles, necessitating more *in vitro* and *in vivo* testing.

# 1 Introduction

Proton pump inhibitors (PPIs) are essential medications recognized by the World Health Organization (WHO), used as a cornerstone of drug therapy for patients with moderate-to-severe Gastroesophageal Reflux Disease (GERD) symptoms and esophagitis [1,2]. PPIs increase gastric pH by covalently binding to the $H^+/K^+$-ATPase antiporter pumps of the gastric parietal cells, inhibiting hydrogen ion discharge into the stomach [3]. PPIs are superior to traditional treatment options like sucralfate, antacids, and histamine-2 receptor antagonists ($H_2$RAs) due to their high effectiveness and rapid increase in gastric pH [4–6]. They have recently replaced $H_2$RAs in treating many acid-related conditions, as they are not associated with the rapid tachyphylaxis seen with $H_2$RAs, the most widely used agents in prophylactic acid suppression [4].

There are currently six Food and Drug Administration (FDA)-approved PPIs: rabeprazole, lansoprazole, pantoprazole, esomeprazole, omeprazole, and dexlansoprazole. These PPIs can effectively treat and prevent conditions like acid reflux, peptic ulcer disease, and first-line treatment for *Helicobacter*-induced gastrointestinal lesions [7], duodenal ulcers, and Zollinger-Ellison syndrome [8,9]. These were sequentially developed due to varying pharmacokinetic parameters, such as extended plasma half-life, routes of administration, and drug interactions [10,11]. Pantoprazole and rabeprazole, used in recommended dosages, maintain esophageal healing and provide symptom relief similar to omeprazole and lansoprazole [12].

In general, the stomach's acids play a significant role in the digestion of nutrients; however, lowering these acids can help with indigestion and heartburn. The stomach produces acidic liquids with a pH value of about 1.5 to 3.5, which kills microorganisms and facilitates the digestion and absorption of nutrients like protein, iron, calcium, and vitamin B12 [13]. However, as this produced acid may harm the digestive system, several protective mechanisms, such as mucosal mucous, bicarbonate barrier, and gastroesophageal junction sphincter contraction, are present to guard against injury to the gastroesophageal junction brought on by gastric secretion [14,15]. When acid secretion overcomes such protective processes, the gastrointestinal mucosa might become injured and inflamed, resulting in unpleasant symptoms or pathological illness [15]. These pathological illnesses, including Barrett's esophagus, gastroduodenal ulcers, GERD, and functional dyspepsia, are known as acid-related diseases [16,17]. After beginning to be used clinically for the treatment of disorders

associated with acid reflux, the usage of PPIs developed consistently and significantly, and now they are among the most often prescribed medications worldwide [18].

Like any other clinical medication, PPIs also come with their own set of benefits and side effects. Compared to H$_2$RAs, long-term use of PPIs is associated with a slightly higher risk of certain side effects [19]. Prolonged use is accumulating evidence of adverse side effects, yet the causal relationship between these symptoms remains unclear. These include a higher chance of getting community-acquired pneumonia, an enteric infection, gastrointestinal tract cancer, malabsorption of many nutrients, vitamins, and minerals, having a myocardial infarction, breaking bones (most often in the hip, spine, and wrist), diarrhoea, a *Clostridium difficile* infection, a stroke, Alzheimer's dementia, and kidney damage [18,20–29]. For patients with fewer side effects, creating a PPI with a longer half-life that will block stomach proton pumps for a more extended period and possibly have a more significant acid suppression effect is crucial. It is important to note that several research teams have been developing potassium-competitive acid blockers and PPIs to investigate longer-lasting and more potent acid inhibition [30–32].

Structural modification of existing drugs can improve their pharmacological properties, potentially leading to more effective and improved therapeutics by rationally altering their molecular structure [33–35]. Such changes can influence drug behaviour by affecting its electrical distribution, polarity, hydrogen bonding capacity, and steric profiles. These alterations influence target binding, solubility, metabolic stability, and bioavailability by altering the molecular composition, weight, and substituent position. This rational design approach enables the development of drug candidates with improved target selectivity, fewer side effects, and the advancement of therapeutic efficacy and safety. This study seeks to develop novel therapeutic alternatives to improve the therapeutic efficacy, potency, and side effects of OMP. Omeprazole was structurally modified and computationally studied by substituting functional groups like –OCH$_3$, –CF$_3$, –OCF$_3$, –NH$_2$, –CH$_2$NH$_2$, –NHCONH$_2$, and –NHCOCH$_3$ for the methoxy groups in the benzimidazole and pyridine rings and the alkyl group in the pyridine ring, resulting in 22 derivatives in this study. The electron-donating or withdrawing effect of functional groups has an impact on the dipole moment, free energy, and the band gap values between the highest occupied molecular orbitals (HOMO) and the lowest unoccupied molecular orbitals (LUMO). The strong electron-withdrawing nature of fluorine leads to an increase in dipole moment in the presence of electronegative atoms or polar bonds [36]. The trifluoromethoxy (–OCF$_3$) group exhibits the largest dipole moment, followed by the acetamido group (–NHCOCH$_3$), which contains a polar carbonyl and an amide group. Through the donation of their lone pairs and the influence of resonance, the methoxy (–OCH$_3$) and amino (–NH$_2$) groups contribute to an increase in the dipole moment. Regarding free energy, electron-donating groups (–OCH$_3$, –NH$_2$) contribute to the stabilization of the molecule, leading to a decrease in free energy; in contrast, electron-withdrawing groups (–OCF$_3$, –NHCOCH$_2$) result in the destabilization of the molecule, thereby increasing free energy [37]. About the HOMO-LUMO gap, the presence of electron-donating groups pushes electro into the system, therefore raising the HOMO level, decreasing the gap, and increasing molecular reactivity, while electron-withdrawing groups increase the gap by pulling electrons from the system, stabilizing the molecule and reducing its reactivity [38]. The presence of the –OCF3 and –NHCOCH3 groups led to an expansion of the HOMO-LUMO gap, thereby enhancing stability. In contrast, the –OCH$_3$ and –NH$_2$ groups contribute to an overall increase in reactivity. These computational chemistry-based modifications of the drug can maximize the pharmacological efficacy, stability, and bioactivity, supporting the development of novel drugs with increased efficacy and reduced side effects. However, more research is necessary to validate and identify a superior drug alternative.

This study provides a detailed *in-silico* analysis of the reference compound OMP and twenty-two of its structural analogues. It evaluates the thermodynamic properties, frontier molecular orbitals, electrostatic potential maps, Fourier transform infrared spectroscopy (FT-IR), and UV–Visible spectral profiles. Furthermore, molecular docking, non-bonding interactions, ADMET characteristics, and PASS predictions were explored. Following these findings, selected analogues were subjected to molecular dynamics simulations to assess their stability and binding behaviour over time. This study

appears to be the first comprehensive computational investigation providing insights into physicochemical, spectral, and biological characteristics while aiding in the rational design of effective, safer therapeutic candidates.

## 2 Methodology and computational details

### 2.1 Geometry optimization

Quantum mechanical methods have significantly aided computational drug design by predicting molecular orbitals, electrostatic potential, and thermodynamic properties [39,40]. The initial geometry of OMP was obtained from the PubChem online database (PubChem CID: 4594). The Gabedit (version 2.5.0) software was used to determine the compounds' most stable and lowest-energy conformation. The structural modification and geometry optimization of all compounds was conducted using the Gaussian 09 W Revision D.01 package, employing CAM-B3LYP and DFT at the 6-31G+ (d, p) level of theory [41,42]. The electronic transitions of the compounds were also determined using the time-dependent density functional theory (TD-DFT) with the same basis set [43]. All the compounds presented in Tables 1, S1 and S1 Fig in S1 File have been analyzed for a range of properties, such as enthalpy, free energy, electrostatic potential, dipole moment, and vibrational frequencies. Frontier molecular orbital features HOMO, and LUMO were calculated at the same theoretical level as follows [44].

$$\text{Gap } (\Delta E) = [\varepsilon \text{LUMO} - \varepsilon \text{HOMO}]$$

(1)

**Table 1. Molecular formula (MF), molecular weight (MW) of OMP, and its newly designed analogues (the remaining ones are presented in S1 and S2 Tables in S1 File).**

| Name | $R_1$ | $R_2$ | $R_3$ | MF | MW (g/mol) |
|---|---|---|---|---|---|
| OMP | $CH_3$ | $OCH_3$ | $OCH_3$ | $C_{17}H_{19}N_3O_3S$ | 345.416 |
| OMP1 | $OCH_3$ | $OCH_3$ | $OCH_3$ | $C_{17}H_{19}N_3O_4S$ | 361.415 |
| OMP2 | $CF_3$ | $OCH_3$ | $OCH_3$ | $C_{17}H_{16}F_3N_3O_3S$ | 399.387 |
| OMP3 | $CH_3$ | $CF_3$ | $OCH_3$ | $C_{17}H_{16}F_3N_3O_2S$ | 383.388 |
| OMP4 | $CH_3$ | $OCH_3$ | $CF_3$ | $C_{17}H_{16}F_3N_3O_2S$ | 383.388 |
| OMP5 | $OCF_3$ | $OCH_3$ | $OCH_3$ | $C_{17}H_{16}F_3N_3O_4S$ | 415.387 |
| OMP6 | $CH_3$ | $OCF_3$ | $OCH_3$ | $C_{17}H_{16}F_3N_3O_3S$ | 399.387 |
| OMP7 | $CH_3$ | $OCH_3$ | $OCF_3$ | $C_{17}H_{16}F_3N_3O_3S$ | 399.387 |
| OMP8 | $NH_2$ | $OCH_3$ | $OCH_3$ | $C_{16}H_{18}N_4O_3S$ | 346.404 |
| OMP9 | $CH_3$ | $NH_2$ | $OCH_3$ | $C_{16}H_{18}N_4O_2S$ | 330.405 |
| OMP10 | $CH_3$ | $OCH_3$ | $NH_2$ | $C_{16}H_{18}N_4O_2S$ | 330.405 |
| OMP11 | $CH_2NH_2$ | $OCH_3$ | $OCH_3$ | $C_{17}H_{20}N_4O_3S$ | 360.431 |
| OMP12 | $CH_3$ | $CH_2NH_2$ | $OCH_3$ | $C_{17}H_{20}N_4O_2S$ | 344.431 |
| OMP13 | $CH_3$ | $OCH_3$ | $CH_2NH_2$ | $C_{17}H_{20}N_4O_2S$ | 344.431 |
| OMP14 | $CONH_2$ | $OCH_3$ | $OCH_3$ | $C_{17}H_{18}N_4O_4S$ | 374.414 |
| OMP15 | $CH_3$ | $CONH_2$ | $OCH_3$ | $C_{17}H_{18}N_4O_3S$ | 358.415 |
| OMP16 | $CH_3$ | $OCH_3$ | $CONH_2$ | $C_{17}H_{18}N_4O_3S$ | 358.415 |
| OMP17 | $NHCONH_2$ | $OCH_3$ | $OCH_3$ | $C_{17}H_{19}N_5O_4S$ | 389.429 |
| OMP18 | $CH_3$ | $NHCONH_2$ | $OCH_3$ | $C_{17}H_{19}N_5O_3S$ | 373.429 |
| OMP19 | $CH_3$ | $OCH_3$ | $NHCONH_2$ | $C_{17}H_{19}N_5O_3S$ | 373.429 |
| OMP20 | $NHCOCH_3$ | $OCH_3$ | $OCH_3$ | $C_{18}H_{20}N_4O_4S$ | 388.440 |
| OMP21 | $CH_3$ | $NHCOCH_3$ | $OCH_3$ | $C_{18}H_{20}N_4O_3S$ | 372.441 |
| OMP22 | $CH_3$ | $OCH_3$ | $NHCOCH_3$ | $C_{18}H_{20}N_4O_3S$ | 372.441 |

$$\eta = \frac{[\varepsilon\text{LUMO} - \varepsilon\text{HOMO}]}{2} \tag{2}$$

$$S = \frac{1}{2\eta} \tag{3}$$

$$\mu = \frac{[\varepsilon\text{LUMO} + \varepsilon\text{HOMO}]}{2} \tag{4}$$

$$\chi = -\frac{[\varepsilon\text{LUMO} + \varepsilon\text{HOMO}]}{2} \tag{5}$$

$$\omega = \frac{\mu^2}{2\eta} \tag{6}$$

were applied for calculating the HOMO-LUMO gap, hardness (η), softness (S), chemical potential (μ), electronegativity (χ), and electrophilicity (ω).

## 2.2 Protein preparation, molecular docking, and interactions

Molecular docking is crucial in drug discovery and computational biology, predicting drug interactions with proteins or binding sites using various algorithms and methodologies developed over the years [45,46]. The target protein for proton pump inhibitors was initially identified as potassium-transporting ATPase alpha chain 1 (PTAAC1) with a molecular weight of 114117.74 Da and Uniprot ID P20648. The protein was predicted using AlphaFold and has a very high model confidence value (pLDDT > 90), which employs novel neural network topologies and training approaches based on geometric, physical, and evolutionary constraints to improve structure prediction accuracy [47]. The protein was a transmembrane protein and a human ATP4A gene product. The 3D crystal structure of PTAAC1 was then obtained using the AlphaFold Protein Structure Database. The protein structure underwent energy minimization using Swiss-PdbViewer (Version 4.1.0) software to reduce weak interatomic interactions within the protein [48]. The PyRx (Version 0.8) software package was used for molecular docking against the energy-minimized PTAAC1 protein, analyzing proteins as macromolecules and optimizing OMP and its analogues as ligands [49]. Docking was carried out after the ligands and proteins were loaded. The software command was used to set the box size to the highest dimension level, with a center grid box size of 112.59 Å, 83.32 Å, and 108.49 Å in the x, y, and z-axis directions, respectively. This enabled the grid box to encase the protein structure entirely. The visual examination of the active site was conducted with BIOVIA Discovery Studio Visualizer 2021. Further, the study utilized molecular docking and nonbonding interaction computations to assess ligands' stability and binding effectiveness within protein binding sites, analyzing docking findings and providing cumulative results. After evaluating different parameters, the selected compounds for simulation were further re-docked and visualized using Maestro v13.5 for docking validation.

## 2.3 ADMET and PASS prediction

Pharmaco-informatics is a vital part of designing novel drugs. During preclinical drug development, it is important to evaluate a drug's chemical absorption, distribution, metabolism, excretion, and toxicity. To predict these properties, the AdmetSAR online server (http://lmmd.ecust.edu.cn/admetsar2/) was utilized [50]. In addition, to forecast biological activity profiles and identify drug-like organic compounds based on their structural formulas, the PASS (Prediction of Activity Spectra for Substances) web server (http://www.way2drug.com/passonline/) was employed [51]. Both evaluations were conducted using SMILES (Simplified Molecular Input Line Entry System) and structural data files to generate the results.

## 2.4 Molecular dynamics simulation analysis

Molecular dynamics (MD) simulation was used to investigate the protein-ligand complex's structural stability under a specific physiological setting. To confirm the stability of the protein-ligand complex, 100 ns MD simulations were used to examine the selected compound's (OMP3, OMP19, OMP21, and OMP as a control) ability to bind the targeted PTAAC1 protein. The Schrödinger suite's Desmond Maestro v13.5 was used to run an MD simulation for 100 ns. The protein preparation wizard was used to pre-process the complex structure after molecular docking investigations of protein-ligand complexes. An orthorhombic periodic boundary box shape with an interval of 10 × 10 × 10 Å³ was employed for each complex simple point-charge (SPC) water model that was utilized to analyze the system to preserve its volume. The solvated system was randomly supplemented with Na⁺ and Cl⁻ ions to keep the salt concentration at 0.15 M. Through the use of the OPLS4 force field, the system was relaxed and mitigated. Lastly, the constant pressure-constant temperature (NPT) ensemble was implemented at 1.01325 bar of pressure and 300.0 K of temperature. After the system had been relaxed for each complex, the final production cycle was executed with an energy of 1.2 and 100 ps recording intervals.

## 3 Results and discussion

### 3.1 Thermodynamics analysis

Thermodynamic analysis, such as Gibbs free energy, enthalpy, and dipole moment, is crucial for comprehending the stability of drug-receptor interactions, molecular behaviour, and binding efficiency [52]. Subtle structural modifications of a compound can significantly impact the thermodynamic properties, influencing the overall pharmacokinetic profile [53,54]. The negative value of free energy can predict the chemical stability, binding affinity, and spontaneity of a reaction, an important criterion for reflecting binding partners' interactions [55,56]. As shown in **Fig 1**, OMP has a free energy of -1447.092 Hartree. In comparison, OMP5 exhibits the largest negative value (-1820.005 Hartree) due to replacing the $-CH_3$ functional group with the $-OCF_3$ functional group in the $R_1$ position of the core structure. The $-OCF_3$ group exhibits a significant electron-withdrawing potential, stabilizes the molecule's electronic structure, and lowers its energy relative to the electron-donating $-CH_3$ group. Regarding free energy, all OMP analogues exhibit larger negative free energy values compared to OMP, except OMP9, OMP10, OMP12, and OMP13, which separately replace the $-OCH_3$ functional group in the $R_2$ and $R_3$ positions with the $-NH_2$ group and the $-CH_2NH_2$ group. Substituting $-OCH_3$ with $-NH_2$ and $-CH_2NH_2$ results in more pronounced electron-donating effects and possible hydrogen-bonding interactions. These changes may stabilize positive charges, improve solubility, or influence reactivity at the $R_2$ and $R_3$ positions, contingent upon the molecular context. The significant negative free energy values of all analogous compounds indicate that these are more stable in terms of energy and configuration. The polarity of a molecule can be determined by measuring the dipole moment. The compounds' polar characteristics and dipole moments contribute to the binding affinity and nonbonding interactions between drugs and receptor proteins in complexes [57,58]. The dipole moment values are influenced by the functional group and its position within analogues. The analysis revealed that OMP17 exhibited the lowest dipole moment values, while OMP18 showed the highest (Fig 1b, S3 Fig in S1 File). The dipole moment of OMP is measured at 4.398 Debye; in contrast, most of its analogues exhibit higher dipole moment values. These analogues' elevated dipole moment values suggest greater binding affinity, hydrogen bonding, and nonbonding interaction capabilities than OMP. Alongside, the highest electrophilicity was observed for OMP2 (3.074 eV), correlating with enhanced reactivity but posing a risk of kinetic instability owing to its low thermodynamic stability. OMP19 and OMP21 emerged as balanced candidates, combining moderate stability and reactivity. Conversely, OMP7, OMP10, OMP18, and OMP22 showed limited drug-likeness owing to excessive polarity (dipole >10) or positive internal energy, whilst OMP8 and OMP9 were deemed unsuitable due to elevated internal energy and reduced stability. All the thermodynamic results, summarized in Fig 1, Tables 1, and 2, highlight critical stability-solubility trade-offs for drug development.

## 3.2 Frontier molecular orbital analysis

The energy gaps between the HOMO and LUMO exhibited by the OMP analogues serve as important indicators of their tendency for electron transfer. This property influences the compound's reactivity, stability, chemical softness, hardness, electrophilicity, and chemical potential values. Therefore, it plays a significant role in the broader context of our research. The control OMP demonstrates a moderate gap, serving as a baseline for assessing the reactivity and stability of the other analogues. The differences observed in HOMO-LUMO gaps across the analogues highlight the influence of

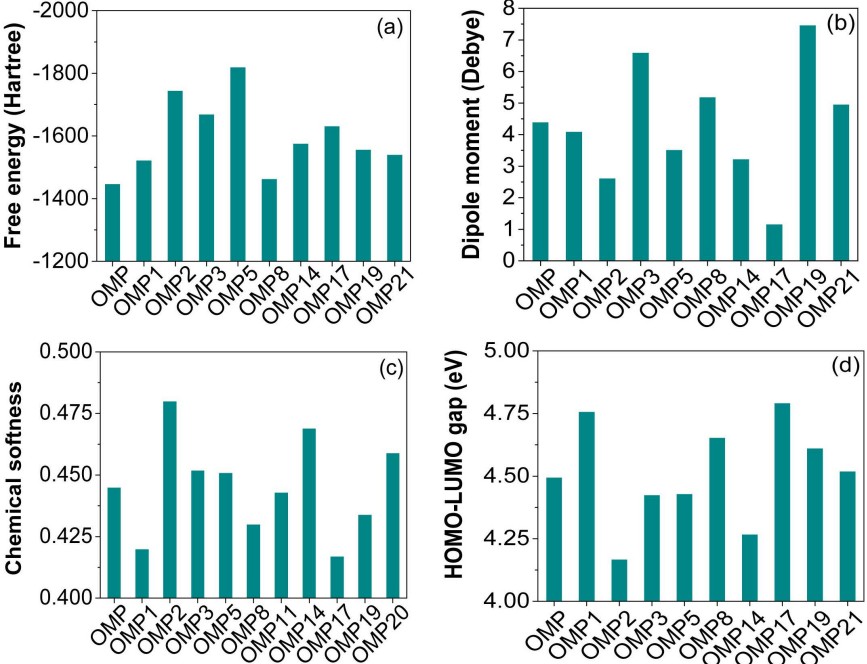

**Fig 1. Free energies (Hartree) (a), dipole moments (Debye) (b), chemical softness (c), and HOMO-LUMO gaps (d) of OMP and a few selected analogues (remaining values are presented in S3 Fig in S1 File).**

**Table 2. Energy (eV) of HOMO, LUMO, hardness (η), chemical potential (μ), electronegativity (χ), and electrophilicity (ω) of OMP and its analogues (remaining are presented in S2 Table in S1 File).**

| Name | ∈HOMO | ∈LUMO | η | μ | χ | ω |
|---|---|---|---|---|---|---|
| OMP | -5.836 | -1.342 | 2.247 | -3.589 | 3.589 | 2.866 |
| OMP1 | -5.855 | -1.097 | 2.379 | -3.476 | 3.476 | 2.539 |
| OMP2 | -5.663 | -1.495 | 2.084 | -3.579 | 3.579 | 3.074 |
| OMP3 | -5.742 | -1.317 | 2.213 | -3.530 | 3.530 | 2.815 |
| OMP5 | -5.563 | -1.133 | 2.215 | -3.348 | 3.348 | 2.531 |
| OMP8 | -5.427 | -0.773 | 2.327 | -3.100 | 3.100 | 2.065 |
| OMP11 | -5.542 | -1.029 | 2.257 | -3.285 | 3.285 | 2.391 |
| OMP14 | -5.619 | -1.351 | 2.134 | -3.485 | 3.485 | 2.846 |
| OMP17 | -5.756 | -0.963 | 2.396 | -3.360 | 3.360 | 2.355 |
| OMP19 | -5.551 | -0.940 | 2.306 | -3.246 | 3.246 | 2.284 |
| OMP20 | -5.431 | -1.073 | 2.179 | -3.252 | 3.252 | 2.426 |
| OMP21 | -5.554 | -1.034 | 2.260 | -3.294 | 3.294 | 2.401 |

different substituents on electronic behaviour. Notably, OMP4 exhibits the highest energy gap of 5.020 eV, suggesting higher energy is required for its excitation. Incorporating a strong electron-withdrawing group like –CF$_3$ into the molecular structure might facilitate the tight binding of electrons while simultaneously reducing their delocalization. This alteration is likely to enhance the overall stability of the compound OMP4 while concurrently reducing its reactivity. In contrast, OMP2 demonstrated the smallest gap of 4.168 eV, underscoring the notable influence of electron-withdrawing groups like –OCH$_3$ in its composition. This, in turn, results in high electrophilic reactivity but potential instability. The nuanced variations in energy gaps among the OMP derivatives, including OMP1, OMP21, OMP11, and others. Compared to OMP, OMP3 exhibits a smaller gap of 4.425 eV, suggesting a likely increase in electronic reactivity due to a possible electron donor effect. Conversely, OMP19 exhibits a larger gap of 4.612 eV compared to the control, indicating enhanced stability and reduced reactivity. OMP21 exhibited a difference of 4.520 eV from the control, suggesting minimal electronic alteration. Among the remaining analogues, the gap values, ranging from 4.300 to 4.800 eV, prove the varying effects and balance substitution of electron-donating and withdrawing group effects, which tune orbital energies. Table 2, Figs 1c and 2 depict the molecular orbitals (HOMO and LUMO) and the HOMO-LUMO energy gap, respectively, of the OMP and a few selected analogues.

### 3.3 Molecular electrostatic potential analysis

Molecular Electrostatic Potential (MEP) map provides a visual representation of charge distribution within a compound, aiding in predicting a drug's reactivity, binding regions, and sites [59]. It is crucial in computational chemistry to identify the probable sites for electrophilic or nucleophilic attack to understand a chemical reaction [60]. This highlights the negatively charged electron-rich region (red colour), and the electron-deficient region, positively charged region (blue colour), allowing the identification of potential sites for hydrogen bonding, ionic interactions, and other noncovalent forces [61,62].

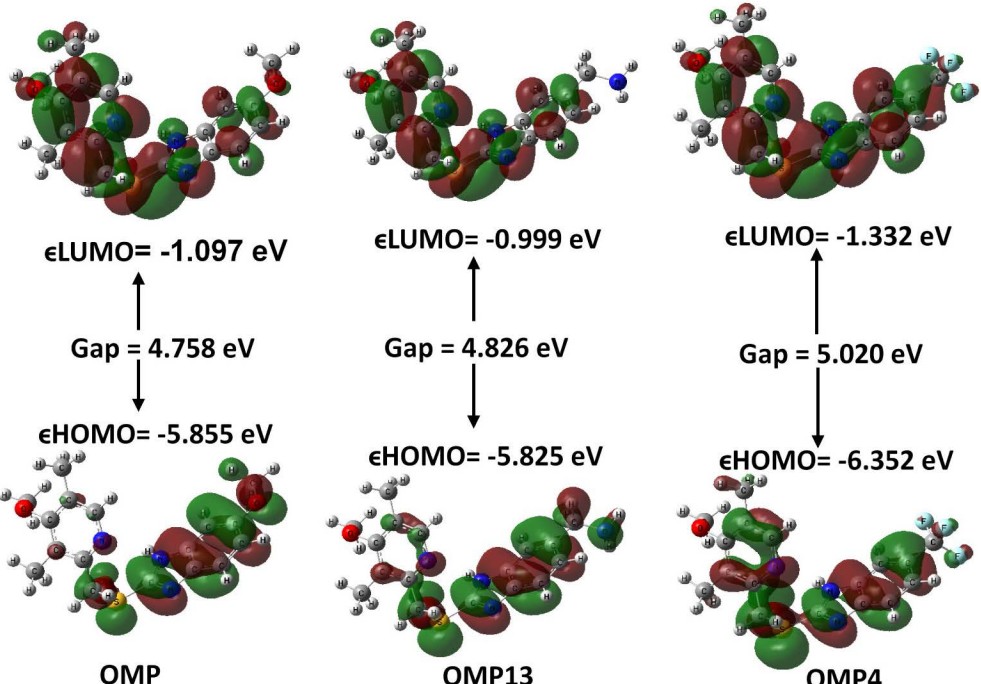

**Fig 2. Molecular orbitals (HOMO and LUMO) and HOMO-LUMO energy gap of OMP and a few selected analogues (remaining values are presented in S2 Fig in S1 File).**

MEP maps facilitate structure-activity relationship studies and binding efficacy by providing comprehensive insights into the electrostatic complementarity between the ligand and its target [59,63]. Our analysis reveals that OMP possesses moderate electrostatic potential values, suggesting a balanced reactivity with electrophilic and nucleophilic tendencies (-0.2339 Hartree and +0.1809 Hartree). Compared to the control, OMP9 and OMP18 exhibit the highest negative potentials of -0.3215 Hartree and -0.3164 Hartree, respectively. This suggests a strong electrophilic characteristic, which could enhance their reactivity toward nucleophilic biological sites. Conversely, OMP3 exhibits the highest positive potential at +0.2740 Hartree, suggesting a pronounced nucleophilic nature, likely favoring interactions with electrophilic targets (Fig 3). Other analogs, such as OMP5 and OMP10, displayed moderate potentials, suggesting well-balanced reactivity profiles that position them as promising candidates for drug development, as they integrate stability with selective reactivity. Overall, the moderate electrostatic potentials of many analogs suggest they offer a promising balance of reactivity for therapeutic applications. At the same time, the variations observed in OMP9, OMP18, and OMP3 simultaneously present opportunities for focusing on more precise interactions.

### 3.4 Vibrational frequencies (FT-IR) analysis

FT-IR spectral analysis is an essential technique for inspecting the chemical structure of any compound, effectively confirming the existence of various functional groups within the molecule. In our study, the FT-IR spectral vibrational frequencies are calculated in the 400-4000 cm⁻¹ range to prove the existence of intended functional groups for OMP and its analogues (Fig 4). The data were then adjusted by multiplying a scaling factor 0.9688 to improve accuracy, as shown in S3 Table in S1 File. The peaks observed at 976-1071 cm⁻¹ attributed to the stretching of the S=O groups confirm the presence of S=O groups in all compounds. All optimized compounds exhibit stretching frequency bands ranging from 3495 to 3575 cm⁻¹ for the N-H bond present in the imidazole units of their structure. Additionally, the stretching frequency was detected within the ranges of 1571-1627 cm⁻¹, 3039-3147 cm⁻¹, and 2895-2935 cm⁻¹, indicating the presence of C=N, aromatic C-H, and aliphatic C-H bonds in all compounds. The N-H vibrations in groups, such as –NH₂, –CH₂NH₂, –CONH₂, –NHCONH₂, and –NHCOCH₃, are observed in the OMP analogues within the 3425-3501 cm⁻¹ range. Furthermore, some analogues have observed at the spectral bands at 1717-1745 cm⁻¹ correspond to the carbonyl group (C=O) present in –CONH₂, –NHCONH₂, and –NHCOCH₃, which are incorporated through modifications. The variation in the stretching frequency position for each was observed concerning the structural modification of the OMP at different positions using different functional groups.

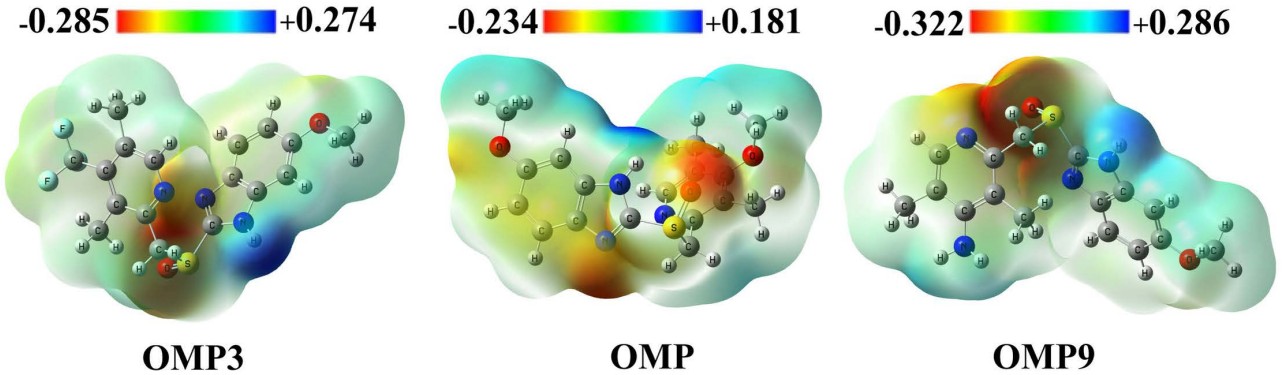

**Fig 3. Electrostatic potential map of OMP, OMP9, and OMP18** (remaining values are presented in S4 Fig in  S1 File).

## 3.5 UV-visible spectral analysis

UV-visible spectroscopy is essential in developing novel drug compounds, offering insights into molecular properties and interactions. This method is crucial for identifying functional groups that influence a drug's biological activity by analyzing the electronic transitions in molecules and revealing information about conjugated systems, aromatic rings, or chromophores [64]. Our analysis focused on assessing the synthesized OMP analogues' absorption properties and drawing comparisons with the OMP to pinpoint modifications that could improve therapeutic efficacy. Pronounced red shifts in $S_0 \rightarrow S_1$ transitions were observed for analogues, like OMP3 ($\lambda_{max}$ = 953.120 nm) and OMP19 ($\lambda_{max}$ = 1130.760 nm), attributed to extended π-conjugation, that can be strategically enhanced through the addition of functional groups. Electron-donating groups (e.g., $-OCH_3$, $-NH_2$) and electron-withdrawing groups (e.g., $-OCF_3$, $-NHCOCH_2$) influence the electron density across π-system, accelerate the intramolecular charge transfer, and improve the delocalization. Such substituents improve conjugation and reduce the HOMO–LUMO gap when positioned effectively, leading to a significant red shift. These structural modifications influence the electronic stability, membrane permeability, and binding affinity towards the target, which are crucial for the rational design of different, more potent candidates for drug development. OMP21 was identified as red shift absorption ($\lambda_{max}$ = 762.640 nm), suggesting the enhanced π-conjugation and moderate excitation probability, making it a potential candidate for increased membrane permeability and prolonged biological activity with $H^+/K^+$-ATPase protein. OMP8 ($\lambda_{max}$ = 695.100 nm) and OMP12 ($\lambda_{max}$ = 678.890 nm) possess absorption values close to OMP ($\lambda_{max}$ = 680.250 nm), indicating their structural similarities and balanced modifications, while slight red shift suggests improved pharmacokinetic properties. Additionally, weak transition probabilities were noted for OMP14, OMP15, OMP18, and OMP22 despite having red shift $\lambda_{max}$ values, suggesting limited electronic excitation. This indicates that further modifications are required to improve these analogues for better conjugation, charge transfer, and overall pharmacokinetic and pharmacodynamic properties. Absorption characteristics, such as the maximum wavelengths, excitation energies, and oscillator strengths for each molecule, were compiled in S4 Table in S1 File. Additionally, Fig 5 visualized the UV-Vis spectra of these molecules.

## 3.6 Binding affinity and interactions analysis

Molecular docking is a crucial computational technique in drug design that optimizes the spatial arrangement of ligands and proteins to anticipate interaction strength, akin to a lock and key mechanism [45,46]. It also contributes to accurate hit detection, lead optimization, and rational drug design, making it a promising option for future drug discovery. The clinical understanding of the binding affinities of the synthesized OMP analogues improved through a comparative analysis of

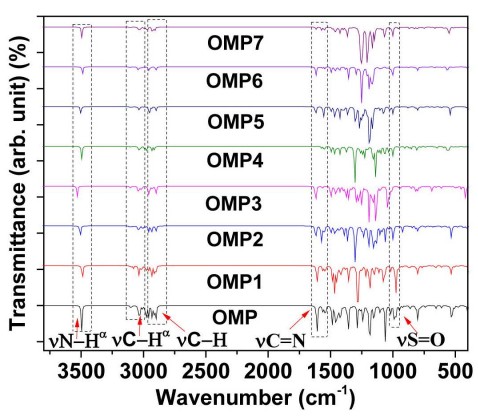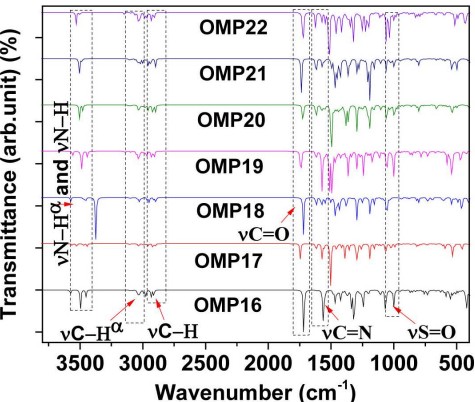

**Fig 4. FT-IR spectra of OMP and its analogues (remaining values are presented in S5 Fig in S1 File).**

their binding energies to those of widely recognized commercial proton pump inhibitors (PPIs), such as pantoprazole, esomeprazole, and omeprazole (control). The calculated binding energies for pantoprazole, esomeprazole, and omeprazole were -7.2 kcal/mol, -7.3 kcal/mol, and -7.0 kcal/mol, respectively, in association with the targeted PTAAC1 protein. An increased binding affinity of several OMP analogues was observed, with binding energies ranging from -6.9 kcal/mol to -8.3 kcal/mol (S7 and S8 Figs, S5 Table in S1 File). In particular, OMP (-7.0 kcal/mol), OMP3 (-7.9 kcal/mol), OMP4 (-7.7 kcal/mol), OMP6 (-7.5 kcal/mol), OMP17 (-7.8 kcal/mol), OMP18 (-7.6 kcal/mol), OMP19 (-8.3 kcal/mol), and OMP21 (-8.1 kcal/mol) exhibited enhanced binding interactions compared to both the control (OMP) and established PPIs. The findings also reveal the presence of multiple binding pockets for analogues and highlight a variety of nonbonding interactions such as alkyl, hydrogen, carbon-hydrogen, conventional hydrogen, hydrophobic bonds, and others which are critical for drug stability and binding affinity in the ligand-protein complex. By changing binding preferences and making preferred ligands more stable, hydrogen and hydrophobic bonds improve the effectiveness of drugs. Strong hydrogen bonding with less than 2.3 Å increases binding affinity, while excellent bond distances are observed in most analogues [39]. The enhanced binding affinities observed in the OMP analogues can be attributed to their distinct molecular interactions with specific amino acid residues located within the binding pocket of the relevant target protein. The non-bonding interactions of OMP (control) were identified to form hydrogen bonds with ASN991 (2.402 Å), TYR801 (2.965 Å), and ASP139 (2.535 Å) as shown in Table 3 and Fig 6. Among the analogues, OMP3 (-7.9 kcal/mol) demonstrated notable interactions with key residues, such as LYS784 (H bond, 2.039 Å), ARG951 (H bond, 2.279 Å), and ASP853 (H bond, 2.687 Å). Moreover, OMP3 interacted with hydrophobic residues like ARG777 and ILE842, stabilizing the binding complex. A comparable trend was noted with OMP19 (−8.3 kcal/mol), which established multiple interactions, carbon-hydrogen bonds with SER380 (2.572 Å), and hydrophobic interactions with LEU378 (2.508 Å) and ILE722 (2.255 Å), thus improving its binding affinity. OMP21 exhibited a binding energy of -8.1 kcal/mol, showcasing notable interactions, including a hydrogen bond with THR381 (2.711 Å), carbon-hydrogen bond with ASP740 (2.103 Å) and GLU376 (2.254 Å), contributed significantly to its binding. The findings suggest that the OMP analogues, especially OMP3, OMP19 and OMP21, exhibit enhanced binding affinities compared to commercially available PPIs, hence selected for further studies. This molecular docking underscores the complex web of nonbonding interactions in the ligand-protein complex (OMP and its analogues with PTAAC1), highlighting their potential contribution to enhanced pharmacological effects, including improved potency and selectivity.

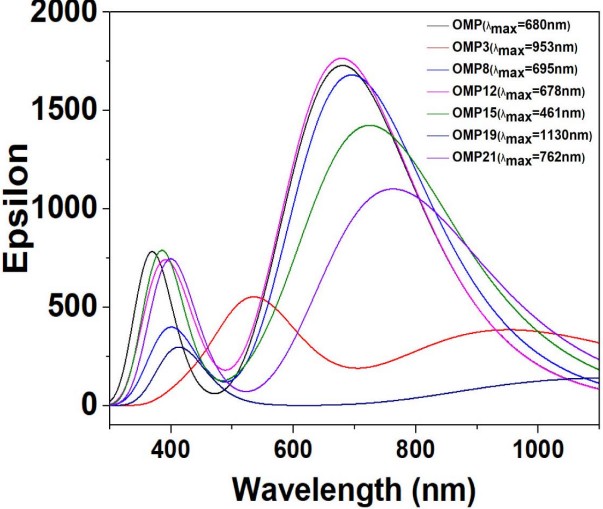

**Fig 5. UV-visible spectra of OMP and a few of its analogues (remaining values are presented in S6 Fig in S1 File).**

**Table 3. Binding affinity and nonbonding interactions of lead and control compounds with the PTAAC1 protein.**

| Name | Binding affinity (kcal/ mol) | Residues in contact | Interaction type | Distance (Å) |
|---|---|---|---|---|
| OMP | -7 | ASN991 | H | 2.40238 |
| | | TYR801 | H | 2.96557 |
| | | ASP139 | H | 2.53565 |
| | | ALA337 | A | 4.01548 |
| | | VAL333 | A | 4.77205 |
| | | LEU923 | A | 5.33114 |
| | | TYR801 | PA | 5.31099 |
| | | PHE919 | PA | 4.84499 |
| OMP3 | -7.9 | LYS784 | H | 2.03935 |
| | | ARG951 | H | 2.27993 |
| | | ASP853 | H | 2.68796 |
| | | ARG777 | X | 3.05064 |
| | | TYR1034 | PSu | 5.89964 |
| | | ARG848 | A | 3.45658 |
| | | ARG777 | A | 4.11721 |
| | | ARG848 | A | 4.30438 |
| | | ARG777 | A | 3.89063 |
| | | ILE842 | A | 4.49137 |
| | | LEU845 | A | 5.08436 |
| | | PHE780 | PA | 4.41439 |
| | | PHE780 | PA | 4.888 |
| OMP19 | -8.3 | LEU378 | C | 2.50852 |
| | | SER380 | C | 2.57251 |
| | | ILE722 | C | 2.25577 |
| | | ILE722 | C | 2.1259 |
| | | ILE722 | PS | 3.60635 |
| | | ILE722 | A | 3.70575 |
| | | PRO294 | PA | 5.27908 |
| | | LEU378 | PA | 4.55256 |
| | | ILE741 | PA | 5.26048 |
| OMP21 | -8.1 | THR381 | H | 2.71146 |
| | | ASP740 | C | 2.1039 |
| | | GLU376 | C | 2.25455 |
| | | ALA374 | A | 3.63098 |
| | | LEU378 | A | 5.36355 |
| | | LEU378 | A | 4.52511 |
| | | ILE722 | A | 4.60832 |
| | | VAL773 | A | 3.95789 |
| | | PRO294 | PA | 4.515 |
| | | ALA724 | PA | 4.34429 |

Here, A = Alkyl, APS = Amide-pi stacked, C = Carbon hydrogen Bond, H = Conventional hydrogen bond, HB = Hydrogen bond, HP = Hydrophobic bond, PA = Pi-alkyl, Pa = Pi-anion, PC = Pi-cation, Pd = Pi-donor, PS = Pi-sigma, PSu = Pi-sulfur, PPS = Pi-Pi stacked, PPTSh = Pi-Pi T-shaped, X = Halogen (Fluorine) bond.

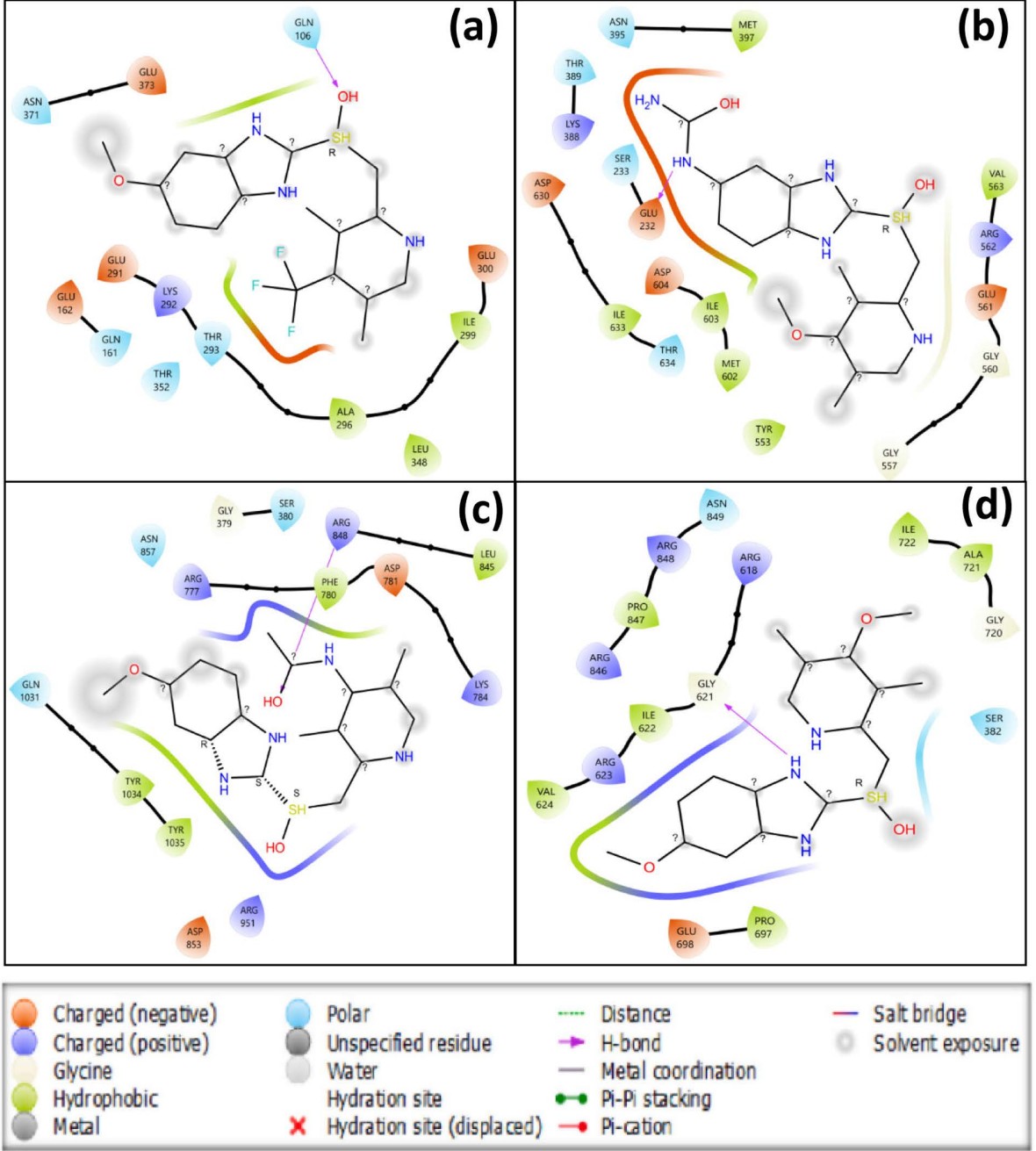

**Fig 6. 2D illustration depicting the interaction between macromolecules (PTAAC1 protein) and ligands (OMP3, OMP19, OMP21, and OMP).**

### 3.7 ADMET prediction

*In-silico* ADMET screening, which includes absorption, distribution, metabolism, excretion, and toxicity, has gained increasing interest as an efficient and cost-effective alternative to *in vivo* drug testing [65,66]. **Table 4** outlines the ADMET properties of OMP and its analogues, which exhibit favourable human intestinal absorption values. However, most of

the analogues demonstrate positive outcomes for the blood-brain barrier (BBB) and could not traverse it like OMP. In addition, all the analogues have been identified as non-carcinogenic and classified as safer in terms of category III acute oral toxicity. Therefore, they pose minimal risks and are suitable for oral consumption. Most OMP analogues can easily pass through the human intestinal CACO-2 cell monolayer assay, which helps determine their permeability and likelihood of being absorbed in the intestines. As reported in the study, the drug experienced efflux back into the intestinal lumen, attributed to the presence of the P-glycoprotein transporter [67]. The drug's bioavailability is inversely affected by P-glycoprotein induction, leading to a reduction, and conversely, its inhibition results in an increase [68]. Further research suggests that suppressing P-glycoprotein could influence a drug's retention, permeability, absorption, and metabolism [69]. In our investigation, OMP and its analogues showed no P-glycoprotein inhibition. These compounds exhibit a low level of oral toxicity in the III category and do not possess carcinogenic properties, indicating that they can be considered relatively safe for oral consumption. The hERG (human ether-a-go-go-related gene) channel, which plays a vital role in cardiac regulation, can be affected by certain medications, potentially resulting in Long QT syndrome and cardiac arrhythmia [70]. The studied OMP analogues demonstrate moderate inhibition, suggesting a favourable safety profile. Around 90% of oxidative metabolic reactions are dependent on the CYP4502C9 iso-enzymes. It is important to mention that all derivatives do not act as substrates for CYP2C9, which reduces the risk of drug-drug interactions [71]. As a result, these analogues are not influenced by CYP2C9 metabolism, which decreases the chances of treatment failures.

## 3.8 PASS prediction

The PASS (Prediction of Activity Spectra for Substances) software predicts over 300 pharmacological effects and biochemical mechanisms by analyzing the structural formula of a substance through a robust SAR analysis of a training set with 30,000 compounds, demonstrating an approximately 86% accuracy in leave-one-out cross-validation [72,73]. From Table 5, it is evident that OMP and its analogues exhibit varying values for different properties. The analogues demonstrated significant effects in various areas, including gastric anti-secretory, anti-ulcerative, and H⁺/K⁺-transporting ATPase inhibition, anti-*Helicobacter pylori*, and CYP1A2 induction. These findings demonstrate that OMP3, OMP10, OMP19, and OMP21 have the most potent gastric anti-secretory and anti-ulcerative effects, respectively. In addition, OMP17 exhibits

**Table 4. Absorption, distribution, metabolism, and toxicological properties (ADMET) studies of OMP and its analogues (remaining are presented in S6 Table in S1 File).**

| Name | Absorption | | | Distribution | | | Metabolism | Toxicity | | | | |
|------|-----|-----|-----|------|------|------|---------|---------|-----------|-----|-----|--------------|
| | HIA | HOB | C2P | BBB | P-GpI | P-GpS | CYP450 2C9 | hERG | Carcinogen | AOT | RAT LD50 | Hepatotoxicity |
| OMP | 0.9968 | 0.8000 | 0.8867 | -0.6326 | NI(0.9680) | NS(0.5573) | NS(0.7838) | WI(0.7190) | NC(0.8318) | III | 2.2254 | 0.6125 |
| OMP1 | 0.9876 | 0.6375 | 0.8180 | -0.7310 | NI(0.9653) | NS(0.5296) | NS(0.7975) | WI(0.8516) | NC(0.8348) | III | 2.3208 | 0.6375 |
| OMP3 | 1.0009 | 0.7857 | 0.8340 | 0.541 | NI(0.9063) | NS(0.5916) | NS(0.7918) | WI(0.9069) | NC(0.8115) | III | 2.2909 | 0.6375 |
| OMP5 | 0.9900 | 0.7714 | 0.7646 | -0.7033 | NI(0.9317) | NS(0.5259) | NS(0.8055) | WI(0.9696) | NC(0.7966) | III | 2.3808 | 0.6375 |
| OMP8 | 0.9830 | 0.7714 | 0.7112 | -0.6573 | NI(0.9674) | NS(0.5562) | NS(0.8459) | WI(0.7679) | NC(0.8050) | III | 2.3486 | 0.6000 |
| OMP11 | 0.9918 | 0.7857 | 0.6358 | -0.6501 | NI(0.9734) | S(0.56710) | NS(0.8609) | WI(0.7823) | NC(0.7711) | III | 2.4562 | 0.5625 |
| OMP14 | 0.9744 | 0.7286 | 0.5000 | -0.8674 | NI(0.9652) | NS(0.5666) | NS(0.8514) | WI(0.9732) | NC(0.7660) | III | 2.4796 | 0.6125 |
| OMP17 | 0.8424 | 0.7857 | -0.5086 | -0.8152 | NI(0.9622) | NS(0.5302) | NS(0.7518) | WI(0.9246) | NC(0.7679) | III | 2.4536 | 0.6250 |
| OMP19 | 0.9236 | 0.7857 | -0.5081 | -0.7538 | NI(0.9476) | NS(0.5610) | NS(0.7123) | WI(0.9418) | NC(0.7795) | III | 2.4625 | 0.6586 |
| OMP20 | 0.9016 | 0.7286 | 0.5197 | -0.8760 | NI(0.8246) | NS(0.5725) | NS(0.7938) | WI(0.9194) | NC(0.7539) | III | 2.4761 | 0.6125 |
| OMP21 | 0.9219 | 0.6429 | 0.5000 | -0.6917 | NI(0.8926) | NS(0.5326) | NS(0.6295) | WI(0.9107) | NC(0.7865) | III | 2.4628 | 0.6625 |

Here, HIA= Human intestinal absorption, HOB= Human oral bioavailability, C2P= CACO-2 permeability, BBB= Blood brain barrier, P-GpI = P-glycoprotein inhibitor, PGpS = P-glycoprotein substrate, hERG = Human ether-a-go-go Related Gene, AOT = Acute oral toxicity, RAT LD50 (mol/kg) = Rat acute toxicity, I = inhibitor, NI = Non-inhibitor, WI = Weak inhibitor, NC = non-carcinogen, NS = Non substrate.

the lowest inhibitory action on the H+/K+-transporting ATPase. Diarrhoea significantly decreases when OMP18 is introduced (from 0.850 to 0.528). In addition, OMP18 has the least impact on hepatitis, stomatitis, and depression compared to the other factors. Based on the findings, it was noted that OMP17, OMP3, OMP18, OMP19 and OMP21 display lower side effects than OMP, while OMP2, OMP4, OMP10, OMP11, OMP13, OMP1 and OMP22 manifests heightened toxicological effects in comparison. The results suggest that specific analogues OMP3, OMP19 and OMP21 demonstrate decreased toxicity compared to the OMP.

### 3.9 Molecular dynamics simulation analysis

To evaluate the stability and dynamic properties of the selected complexes, protein root mean square deviation (RMSD), ligand RMSD, root mean square fluctuation (RMSF), radius of gyration (rGyr), solvent accessible surface area (SASA), polar surface area (PolSA), molecular surface area (MolSA), protein-ligand interaction, and ligand-protein interaction were analyzed using MD simulation trajectory in simulation interaction diagram (SID) (Table 6). Then, hydrogen bond analysis was conducted using simulation event analysis (SEA), and post-simulation MM-GBSA analysis was calculated using the Prime MM-GBSA v3.0 of the Maestro v13.5. PCA was computed from the MD trajectories using the Bio3D package of R programming.

**3.9.1 Protein root mean square deviation (RMSD) analysis.** In MD simulations, RMSD is an important statistic for measuring the average deviation of atom locations over time in relation to a reference structure [74,75]. It assesses structural convergence, similarity, and stability across a 100 ns timeframe by examining specific protein constituents such as side chains, heavy atoms, backbones, and alpha carbons. RMSD also calculates ligand atom deviations when matching trajectory structures with the beginning reference over time, up to 100 ns [76]. Consequently, to examine the target protein's conformational change in the complex of the selected compound (OMP3, OMP19, OMP21) and control (OMP), as depicted in Fig 7a accordingly. After complexing to the apo-protein (PTAAC1), the RMSD average values for the selected compounds, including OMP3, OMP19, OMP21 and OMP, are 6.61 Å, 7.79 Å, 7.31 Å, and 5.29 Å, respectively, out of a total of 1001 frames. The highest RMSD values among the chosen compounds OMP3, OMP19, OMP21, and OMP are as follows: 7.834 Å, 9.085, 9.616 Å and 7.056 Å at frame numbers 855, 877, 616, and 746, respectively out of 1001. However, the lowest RMSD values and frame numbers were 1.957 Å, 2.369, 2.225 in the first frame, and 2.184 Å in the second frame for the selected compounds OMP3, OMP19, OMP, and OMP21, respectively. The compounds (OMP3 and OMP19), when compared to the control (OMP) in Fig 7a, demonstrated better stability with

**Table 5. PASS predicted data of OMP and its analogues (Pa values, indicating to be active) (remaining are presented in S7 Table in S1 File).**

| Name | Gastric Anti-secretory | Anti-ulcerative | H+/K+-transporting ATPase inhibitor | Anti-Helicobacter Pylori | CYP1A2 inducer | Nephritis | Hepati-tis | Stoma-titis | Depres-sion | Diar-rhea |
|------|------|------|------|------|------|------|------|------|------|------|
| OMP | 0.969 | 0.942 | 0.922 | 0.937 | 0.888 | 0.842 | 0.847 | 0.864 | 0.842 | 0.850 |
| OMP1 | 0.823 | 0.892 | 0.893 | 0.893 | 0.940 | 0.851 | 0.800 | 0.833 | 0.902 | 0.820 |
| OMP2 | 0.910 | 0.874 | 0.884 | 0.828 | 0.955 | 0.897 | 0.908 | 0.911 | 0.959 | 0.892 |
| OMP3 | 0.700 | 0.767 | 0.734 | 0.670 | 0.872 | 0.830 | 0.711 | 0.839 | 0.904 | 0.709 |
| OMP5 | 0.717 | 0.857 | 0.836 | 0.900 | 0.843 | 0.881 | 0.783 | 0.857 | 0.722 | 0.546 |
| OMP8 | 0.777 | 0.838 | 0.767 | 0.771 | 0.907 | 0.861 | 0.846 | 0.862 | 0.924 | 0.846 |
| OMP11 | 0.852 | 0.906 | 0.726 | 0.816 | 0.933 | 0.852 | 0.866 | 0.896 | 0.911 | 0.858 |
| OMP14 | 0.864 | 0.872 | 0.738 | 0.807 | 0.950 | 0.891 | 0.888 | 0.914 | 0.958 | 0.862 |
| OMP17 | 0.841 | 0.828 | 0.476 | 0.740 | 0.869 | 0.771 | 0.621 | 0.687 | 0.737 | 0.607 |
| OMP19 | 0.971 | 0.930 | 0.709 | 0.797 | 0.935 | 0.858 | 0.834 | 0.880 | 0.817 | 0.837 |
| OMP20 | 0.736 | 0.825 | 0.524 | 0.851 | 0.958 | 0.864 | 0.780 | 0.735 | 0.802 | 0.820 |
| OMP21 | 0.903 | 0.826 | 0.559 | 0.638 | 0.707 | 0.735 | 0.626 | 0.721 | 0.698 | 0.566 |

a slightly higher fluctuation at a simulation time range of 0 ns to 15 ns and maintained constant fluctuation with minimal conformational changes between 15 ns to 100 ns. On the other hand, during the simulation time range of 0-70 ns and 0-50 ns, the compound OMP21 and OMP showed comparatively high fluctuation and maintained the lowest continuous fluctuation from 70-100 ns and 50-100 ns, respectively.

**3.9.2  Ligand RMSD analysis.** The Ligand RMSD is a useful method for evaluating conformational changes and ligand stability inside a binding pocket, and RMSD fluctuations indicate possible destabilization or conformational changes [75,77]. The complex docking structure was first aligned with the reference protein backbone to determine the RMSD of the selected compounds OMP3, OMP19, OMP21, and OMP in Fig 7b. The average ligand RMSD values for these compounds, OMP3, OMP19, OMP21, and OMP, were 2.26 Å, 0.96 Å, 0.97 Å, and 1.52 Å, respectively. In the analysis of the chosen compounds, OMP3, OMP19, OMP21, and OMP exhibited lower ligand RMSD values of 0.85, 0.39, 0.249, and 0.514 at frame numbers 4, 120, 1, and 1, respectively. The highest possible values were 2.596, 1.99, 2.731, and 2.799 at frames 505, 12, 849, and 995, respectively. OMP19 and OMP21 had lower average ligand RMSD values than the control, indicating that they are more stable in binding, structural integrity, and prediction accuracy.

**3.9.3  Root mean square fluctuation (RMSF) analysis.** The RMSF values measure the local variations and stability of amino acid (AA) residues in a complex system [77]. Therefore, as shown in Fig 7c, the RMSF values of the compounds OMP3, OMP19, OMP21, and OMP in the complex were assessed to ascertain how protein structural flexibility changes when particular compounds attach to a particular residual position. Specifically, the average RMSF values for OMP3, OMP19, OMP21, and control OMP are 2.2 Å, 2.01 Å, 2.4 Å, and 2.3 Å, respectively. These values are concentrated around 2-3 Å [78], indicating the different levels of flexibility in different regions of the protein structure. The selected three

**Table 6. Using a 100 ns molecular dynamics simulation, three lead compounds (OMP3, OMP19, OMP21) and the control drug (OMP) produced varying parameters, including the highest (H), lowest (L), and average (A).**

| Parameter | Value | OMP | OMP3 | OMP19 | OMP21 |
|---|---|---|---|---|---|
| Protein Cα RMSD | H. RMSD (Å) | 7.056 | 7.834 | 9.085 | 9.616 |
| | L. RMSD (Å) | 2.225 | 1.957 | 2.369 | 2.184 |
| | A. RMSD (Å) | 5.290 | 6.610 | 7.790 | 7.310 |
| Ligand RMSD | H. L-RMSD (Å) | 2.799 | 2.596 | 1.990 | 2.731 |
| | L. L-RMSD (Å) | 0.514 | 0.850 | 0.390 | 0.249 |
| | A. L-RMSD (Å) | 1.520 | 2.260 | 0.960 | 0.970 |
| Protein Cα RMSF | H. RMSF (Å) | 18.480 | 11.08 | 16.461 | 26.762 |
| | L. RMSF (Å) | 0.875 | 0.827 | 0.775 | 1.051 |
| | A. RMSF (Å) | 2.300 | 2.200 | 2.010 | 2.400 |
| Radius of gyration | H. rGyr (Å) | 4.763 | 4.950 | 5.225 | 5.142 |
| | L. rGyr (Å) | 3.686 | 4.111 | 4.535 | 3.858 |
| | A. rGyr (Å) | 4.470 | 4.730 | 4.990 | 4.830 |
| Solvent-accessible surface area (SASA) | H. SASA(Å²) | 515.429 | 215.938 | 417.039 | 237.023 |
| | L. SASA (Å²) | 166.864 | 26.003 | 145.449 | 68.267 |
| | A. SASA (Å²) | 352.090 | 89.670 | 279.42 | 144.160 |
| Molecular surface area (MolSA) | H. MolSA (Å²) | 337.378 | 338.173 | 354.792 | 363.022 |
| | L. MolSA (Å²) | 313.131 | 324.639 | 334.055 | 335.158 |
| | A. MolSA (Å²) | 329.400 | 332.070 | 346.370 | 354.450 |
| Polar surface area (PolSA) | H. PolSA (Å²) | 124.205 | 111.134 | 232.288 | 156.923 |
| | L. PolSA (Å²) | 82.502 | 81.661 | 198.183 | 110.524 |
| | A. PolSA (Å²) | 105.450 | 96.450 | 218.500 | 139.630 |

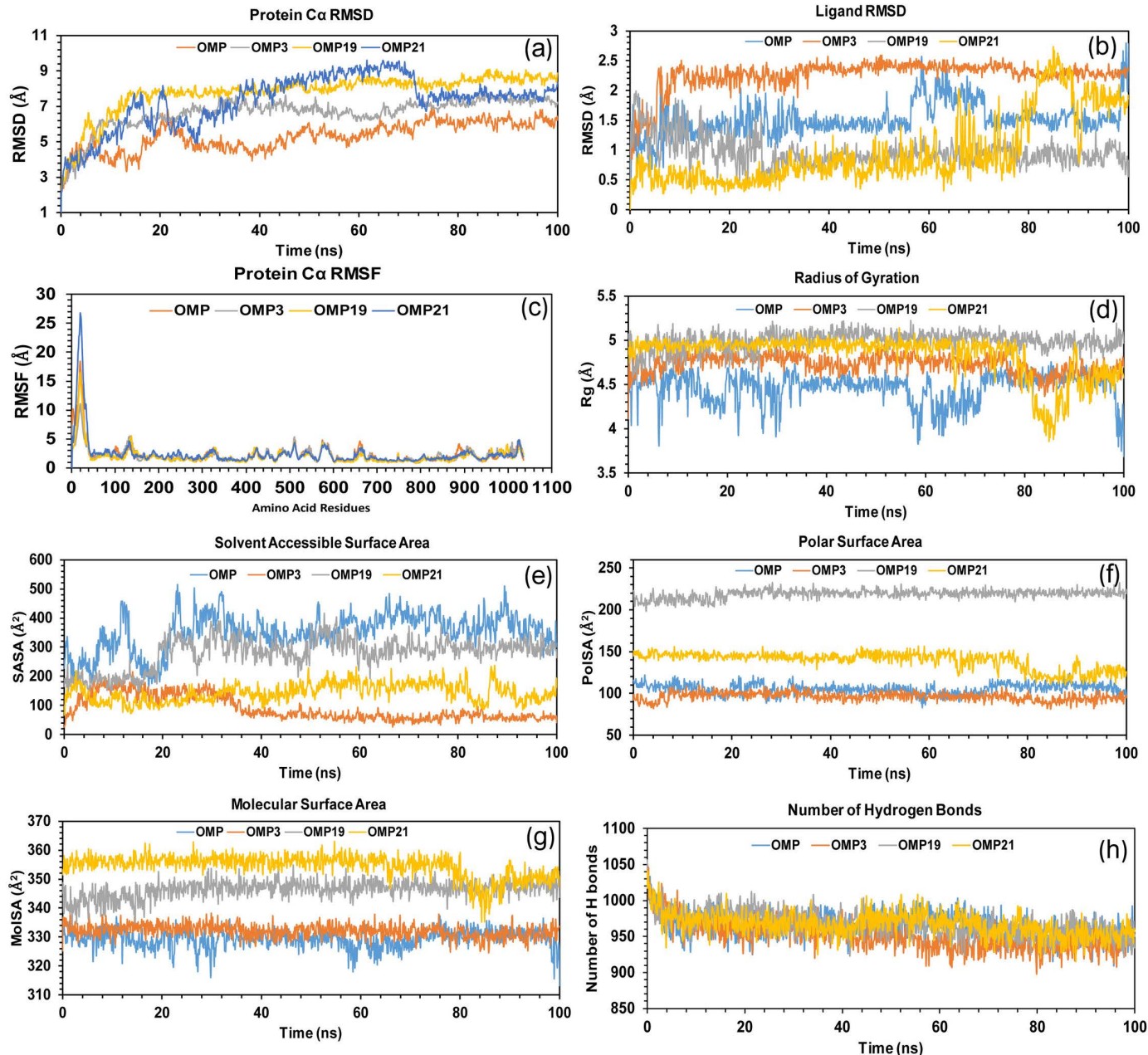

**Fig 7. The protein RMSD (a), Ligand RMSD (b), RMSF (c), rGyr (d), SASA (e), PoISA (f), MoISA (g), and Hydrogen bond (h) analyses are displayed for three selected compounds (OMP3, OMP19, OMP21) and OMP (control); derived from the MD simulation.**

compounds OMP3, OMP19, OMP21, and control OMP had the following RMSF values: the lowest values were 0.827 Å, 0.775 Å, 1.051 Å, and 0.875 Å, respectively, at residual positions LYS793, TYR789, and MET602; the highest values were 11.08 Å, 16.461 Å, 26.762 Å, and 18.48 Å, respectively, at residual positions GLY19. Fig 7c shows a peak region of the protein for each of the chosen compounds. During the simulation period, these residue positions-which included GLY133, ARG510, LYS574, LEU659, and ARG1022 exhibited the most fluctuations. Based on the RMSF, it was discovered that the

stiffest secondary structural elements, like alpha-helices and beta-sheets, have at least 150 to 500 AA and 600 to 1040 AA residues.

### 3.9.4 Radius of gyration (rGyr) analysis.

The radius of gyration (rGyr) measures the root mean square distance of a molecule's atoms from its center of mass and is used in MD simulations to quantify the compactness or overall shape of a molecule, especially proteins and chemical compounds. Throughout the 100 ns simulation period, the stability of the chosen compounds OMP3, OMP19, OMP21, and OMP interacting with the target protein was assessed using rGyr in Fig 7d. The minimum rGyr values recorded for OMP3, OMP19, OMP21, and OMP were 4.111 Å, 4.535 Å, 3.858 Å, and 3.686 Å at frame numbers 1, 17, 849, and 1000, respectively. Conversely, the peak values recorded were 4.95 Å, 5.225 Å, 5.142 Å, and 4.763 Å at frame numbers 517, 570, 547, and 771, as well. Additionally, the average rGyr values for OMP3, OMP19, OMP21, and OMP were found to be 4.73 Å, 4.99 Å, 4.83 Å, and 4.47 Å, respectively. Fig 7d depicts the OMP as a compact and sturdy structure, serving as a benchmark for comparison. The OMP3 curve closely resembles the OMP, with somewhat higher rGyr values while retaining a similar level of compactness and stability, implying minimal structural expansion. Similarly, the OMP19 curve exhibits a slight stabilization above the OMP3 and OMP, indicating a moderate level of compactness with only minimal variations from the OMP. The OMP21 curve shows the highest rGyr values, indicating a bit less compactness and greater flexibility; nonetheless, the overall trend stays pretty aligned with the control. The findings suggest that the structural compactness of our systems aligns closely with the OMP, exhibiting only minor variations that remain within the bounds of benchmark stability.

### 3.9.5 Solvent accessible surface area (SASA) analysis.

SASA is the surface area where the solvent molecules and the protein or ligand come into contact. The simulation analysis shows a correlation with the solvent-complex interactions [79]. The SASA value determines the magnitude and significance of the protein arrangement changes that occur when a ligand attaches to a receptor [80]. A greater SASA value indicates a larger protein volume, and minimal variation is expected during the simulation time [81]. The SASA values for OMP3, OMP19, OMP21, and OMP are determined in Fig 7e. The minimum values recorded for these compounds were 26.003 Å², 145.449 Å², 68.267 Å², and 166.864 Å² in frames 1, 50, 1, and 172, respectively. The highest values for OMP3, OMP19, OMP21, and OMP compounds were 215.938 Å², 417.039 Å², 237.023 Å², and 515.429 Å² in frame numbers 139, 528, 874, and 230, respectively. In frame numbers 139, 528, 874, and 230, OMP3, OMP19, OMP21, and OMP molecules had the greatest values of 215.938 Å², 417.039 Å², 237.023 Å², and 515.429 Å², respectively. The average SASA values for OMP3, OMP19, OMP21, and OMP were 89.67 Å², 279.42 Å², 144.16 Å², and 352.09 Å², respectively. The complicated mechanism had an average SASA value ranging from 85 to 355 Å², indicating that an AA residue was easily accessible to the molecule. The OMP has the greatest SASA values (300–500 Å²). This means that it is the most solvent-exposed and contains fewer interactions or structures. The other compounds (OMP3, OMP19, and OMP21) have substantially lower SASA values, showing that they are either more compact or have stronger interactions to limit surface exposure. OMP19 has SASA values that are slightly reduced compared to OMP (200–400 Å²); in contrast, OMP21 and OMP3 consistently demonstrate the lowest SASA, remaining below 200 Å², which suggests reduced exposure. This indicates that the OMP3, OMP19, and OMP21 molecules exhibit a greater binding potential or enhanced structural stability than OMP.

### 3.9.6 Polar surface area (PolSA) and molecular surface area (MolSA) analyses.

The PolSA and MolSA tests reveal that the chosen compounds; OMP3, OMP19, OMP21, and OMP maintain their stable surface properties during the 100 ns simulation. OMP3 has a slightly higher PolSA of 111.134 Å² and the highest MolSA of 338.173 Å², indicating strong polar surface exposure. OMP19's high PolSA (198-235 Å²) and MolSA (354.792 Å²) indicate a larger surface area. OMP21 exhibits a high PolSA of 110-156 Å² and MolSA of 363.022 Å², indicating a compact structure with a low polar surface area. The simulation showed minimal MolSA and PolSA variations for all systems. This demonstrated the structure's stability and the surface's constant exposure without any significant shape changes. OMP19 and OMP21 exhibit the highest average PolSA (218.50 Å² and 139.63 Å²) and the largest average MSA (346.37 Å² and 354.45 Å²) compared to

OMP (PolSA at 105.45 Å² and MolSA at 329.40 Å²), indicating extensive surface exposure in Fig 7f and 7g. OMP3 has a less compact structure than other compounds, with an average MolSA of 332.08 Å² and a lower PolSA of 96.45 Å². All systems had consistent MolSA and PolSA values, indicating structural stability throughout the experiment. The simulation revealed that all systems had modest MolSA and PolSA variances.

**3.9.7 Hydrogen bond analysis.** Hydrogen bonds (HB), a form of noncovalent contact, substantially influence biomolecule behaviour and three-dimensional structure [82]. The biological recognition process depends on hydrogen bonding, and the molecular dynamics required for biological processes are inextricably linked to the rapid production and breakdown of hydrogen bonds [83]. Hydrogen bonds are required to stabilize the ligand with the target protein, aid in adsorption, accelerate metabolic processes, and improve therapeutic selectivity [76,84,85]. Fig 7h depicts the monitoring of H-bond formation in the protein-ligand complex interaction over a 100 ns simulated period. Consequently, Fig 7h illustrates the number of H-bonds established throughout the interaction of the protein-ligand complex during the 100 ns simulation. The average H-bond numbers for the selected compounds, which include OMP3, OMP19, OMP21, and OMP, were found to be 950.69, 966.88, 965.63, and 964.20, respectively. Throughout the simulation, each compound formed hydrogen bonds ranging from 895 to 1020 simultaneously. The connection between the ligand and receptor will experience significant enhancement and stabilization.

**3.9.8 Protein-ligand and ligand-protein contact analysis.** We used the simulated interactions diagram (SID) to study the complex structure of a protein, its associated ligands, and their molecular interactions over a 100-ns simulation period. In Fig 8, several characteristics, such as H-bonds, ionic, hydrophobic, and water bridge bonds, were used to investigate and demonstrate the interactions between the protein and selected compounds. During the 100 ns simulation, the interactions between each molecule resulted in a stable binding to the target macromolecule. More than two contacts were found in OMP3, with interaction fractions of 0.20, 1.90, 0.28, 0.12, 0.11, and 1.10 for residues LYS31, ARG105, GLN106, GLU162, LYS292, and GLU373, in that order. The particular connection consistently results from the same subtype repeatedly engaging with the ligand throughout the simulation, as shown in Fig 8a. OMP19 had substantial interactions with GLU234 (0.28), GLU653 (1.5), VAL663 (0.55), and ARG668 (0.1), which occurred at optimal times throughout the simulation (Fig 8b). The compound OMP21 established several interactions with the residues ASP853 (0.3), ASN857 (0.2), ARG951 (1.3), TYR1034 (0.8), and TYR10354 (0.1), and the simulation duration effectively preserved these connections (Fig 8c). The simulation time for compound OMP effectively captured various interactions at residues LYS669 (0.2), ASP670 (1.35), ARG694 (0.05), ARG846 (0.1), and ASN (0.03) (Fig 8d). Nonetheless, OMP3 demonstrates a notable interaction involving hydrogen and other bonds with the PTAAC1, as illustrated in Fig 9a.

The simulated interactions diagram (SID) illustrates the protein interactions among the chosen compounds, OMP3, OMP19, OMP21, and OMP, in Fig 9. In contrast to OMP, OMP3 and OMP21 exhibited significant interaction residues (exceeding two) throughout the simulation. This led to several interactions with the identical subtype of ligand, thus maintaining the distinct interaction. The study demonstrated enhanced stability in the interaction between the ligand and protein.

**3.9.10 Principal component analysis (PCA) analysis.** Principal component analysis is a method for reducing dimensionality that might be useful in studying protein structures and dynamics. The protein-ligand complexes' conformational dynamics were examined using Principal Component Analysis (PCA) applied to the trajectories obtained from molecular dynamics (MD) simulations. Fig 10 shows the principal component analysis results for the OMP3, OMP19, and OMP21 complexes, as well as the OMP complex. The first three PCs accounted for a significant portion of the OMP3 complex's overall mobility (Fig 10a). Of the total variation, PC1 was responsible for 36.19 %, PC2 for 16.19 %, and PC3 for 12.8%. Following PC1, the eigenvalue plot displayed a steep decline, indicating that the primary motions were primarily caused by the first principal component. In Fig 10b, we can see that the OMP19 complex showed more conformational variation along PC1, which explained 45.18 % of the total variance,

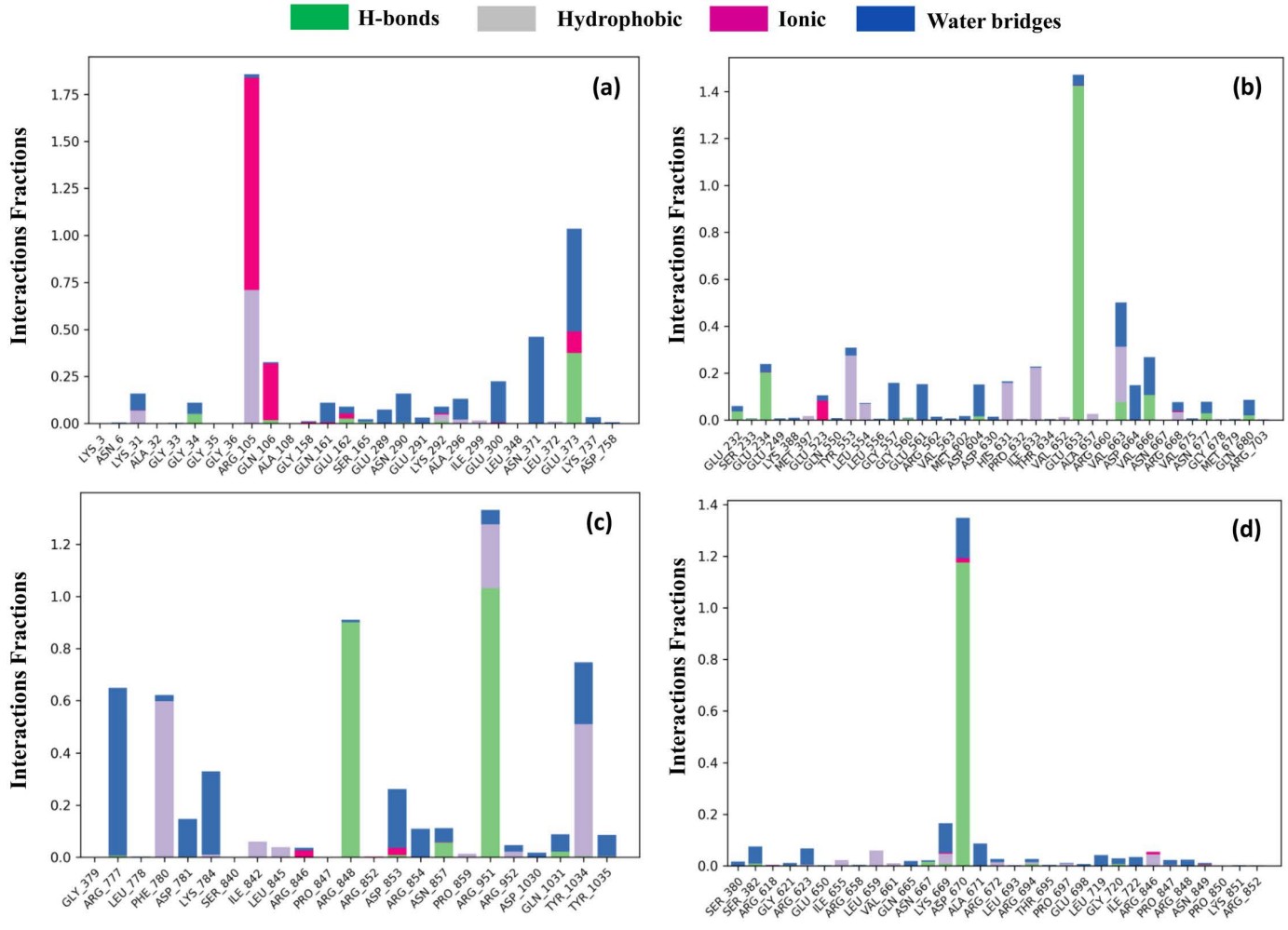

**Fig 8.** The grouped bar charts show the protein-ligand interactions identified during the 100 ns simulation of PTAAC1 protein binding with OMP3 (a), OMP19 (b), OMP21 (c) and OMP (control) (d).

compared to 14.77 % along PC2 and 7.55 % along PC3. In comparison to OMP3, OMP19's eigenvalue plot showed a more gradual decline, indicating somewhat more dispersed dynamic behaviour over several modes. Fig 10c shows the OMP21 complex, where the first principal component (PC1) accounted for almost half of the overall motion, explaining 50.65% of the variance. Two-thirds of the variation was explained by PC2, while the remaining 6.51% was explained by PC3. For OMP21, the more precipitous decline in eigenvalues following PC1 suggests a more consistent and dominating motion along a single mode. Within the control OMP complex (Fig 10d), PC1 accounted for 45.18% of the variance, PC2 for 16.94%, and PC3 for 12.04%. The eigenvalue distribution showed that OMP3 and OMP19 were behaving in an intermediate manner with regard to motion spread. The comparison's findings showed that the OMP21 complex exhibited the most consistent dynamic behaviour. The fact that conformations were more closely packed along the main components and that PC1 recorded the highest variance—50.65%—supported this. OMP19's movement was more widely distributed among numerous PCs, suggesting greater flexibility, whereas OMP3's movement was more spread throughout the initial three PCs. The OMP complex showed moderate stability but somewhat greater dispersion when compared to OMP21. PC1 typically exhibits the most significant variance,

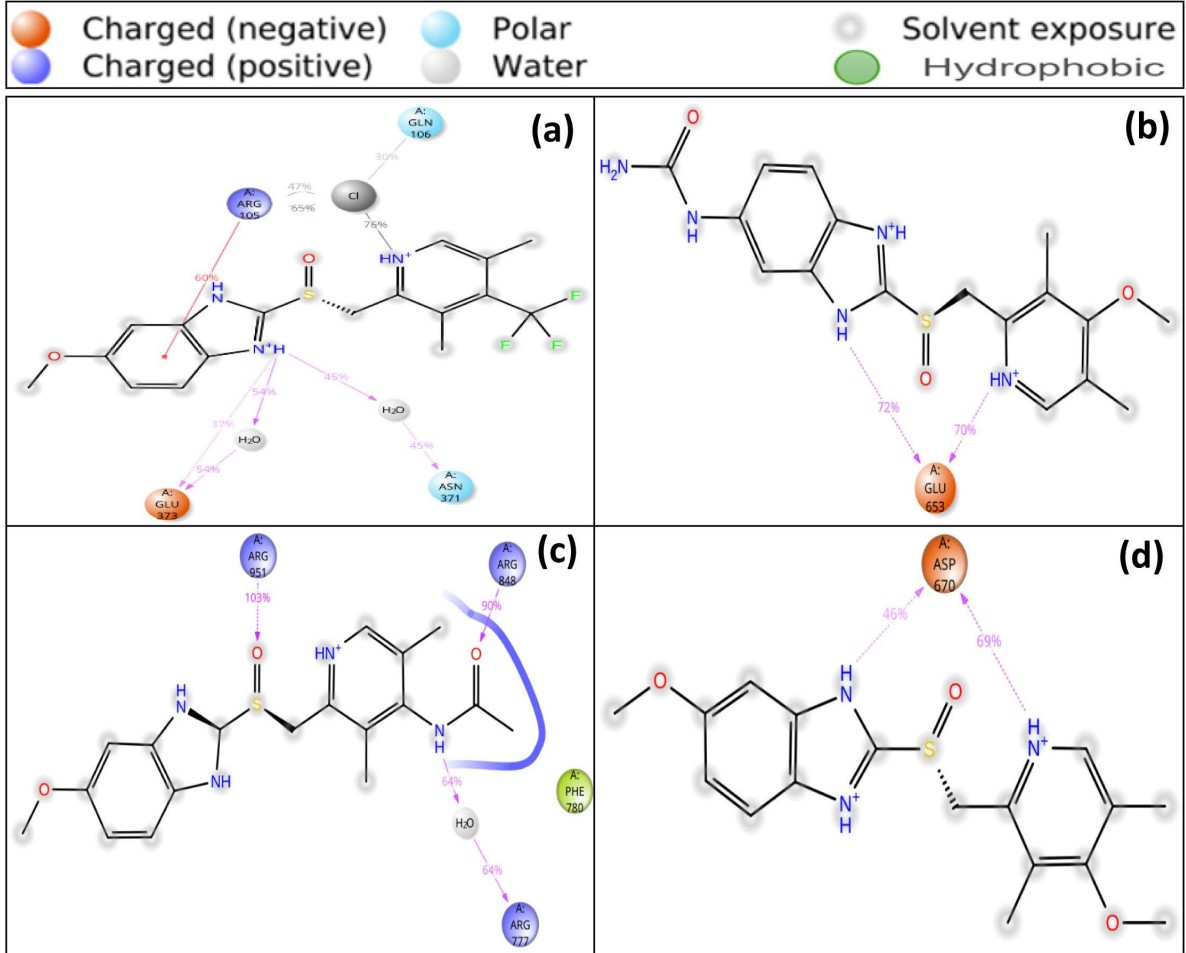

**Fig 9. Analysis of protein-ligand interactions displayed from a 100 ns simulation of the PTAAC1 protein binding with OMP3 (a), OMP19 (b), OMP21 (c) and OMP (control) (d).**

with a reduction in variation observed as we progress through the principal components, suggesting more restricted and localized movements. Distinct conformational variations were observed through a simple clustering approach in the PC subspace. The colours red and white represent the lowest and moderate levels of movement, respectively, whereas blue signifies the highest degree.

**3.9.11 Post-simulation MM-GBSA analysis.** MM-GBSA techniques have been used in this study to compute the binding free energy of the protein of interest in complex with the ligand. Following each 50 ns molecular dynamics simulation, the 100 ns dynamic simulation trajectory was utilized to compute the MM-GBSA for the protein-ligand complex structure. Upon complexation with the targeted protein, the initial (0 ns), intermediate (50 ns), and final (100 ns) negative binding free energy (dG Bind) values for OMP3, OMP19, OMP21, and OMP were recorded as follows: -26.67, -27.74, and -36.91 kcal/mol; -39.67, -22.10, and -26.45 kcal/mol; -64.67, -46.96, and -12.61 kcal/mol; -25.76, -19.87, and -15.02 kcal/mol, respectively (Table 7). Therefore, the selected compounds should be able to bind to the target protein for an extended period.

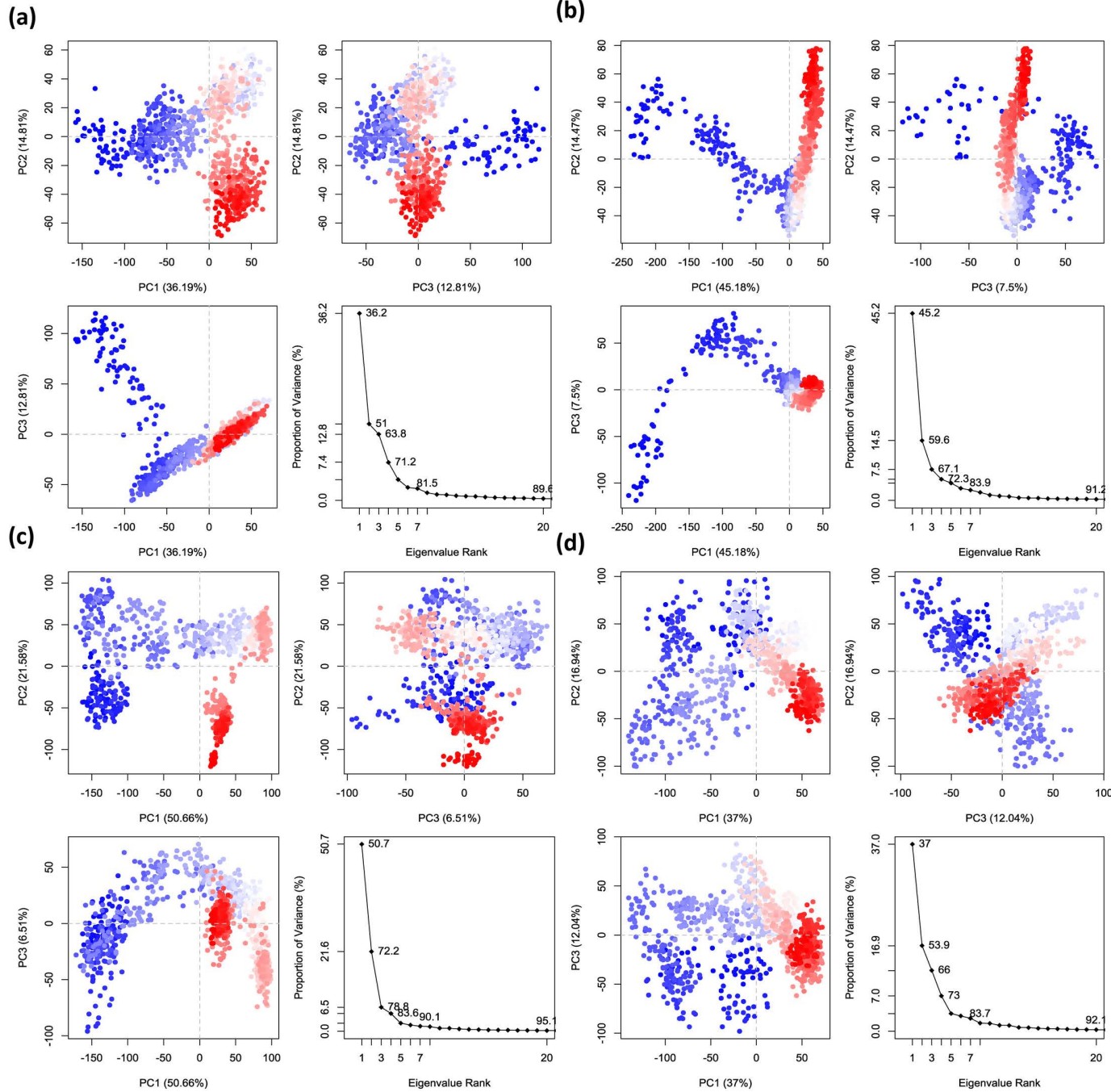

**Fig 10. Eigenvalues in principal component analysis vs proportion of variance.** Each region is displayed on three independent panels. There are three different versions: PC1, PC2, and PC3. In this scenario, the compounds are OMP3 (a), OMP19 (b), OMP21 (c), and OMP (control) (d).

**Table 7. The table depicts the MM-GBSA binding energy components for OMP3, OMP19, OMP21, and OMP (control) with the target protein at initial, intermediate, and final production.**

| Time | dG Bind | dG Bind Coulomb | dG Bind Covalent | dG Bind Hbond | dG Bind Lipophilicity | dG Bind Packing | dG Bind Solvation Generalized Born | dG Bind Van Der Waals |
|---|---|---|---|---|---|---|---|---|
| **MM-GBSA of OMP** | | | | | | | | |
| **0 ns** | -25.760 | 90.724 | 3.046 | -0.100 | -6.197 | -1.178 | -81.304 | -30.751 |
| **50 ns** | -19.875 | 4.292 | 2.080 | -0.547 | -4.917 | 0.000 | -1.080 | -19.703 |
| **100 ns** | -15.024 | 1.934 | 0.671 | -0.201 | -3.170 | -2.548 | 0.992 | -12.704 |
| **MM-GBSA of OMP3** | | | | | | | | |
| **0 ns** | -26.667 | -9.238 | 1.493 | -0.478 | -8.253 | -5.974 | 46.176 | -50.394 |
| **50 ns** | -27.742 | -7.345 | 2.538 | -0.998 | -8.175 | -7.418 | 39.571 | -45.915 |
| **100 ns** | -36.914 | -11.680 | 3.279 | -1.116 | -8.111 | -7.476 | 36.033 | -47.842 |
| **MM-GBSA of OMP19** | | | | | | | | |
| **0 ns** | -39.672 | 9.466 | 2.503 | -3.906 | -9.055 | -0.069 | 6.059 | -44.671 |
| **50 ns** | -22.104 | 19.073 | 0.398 | -1.797 | -2.972 | -3.873 | -11.414 | -21.519 |
| **100 ns** | -26.449 | -1.118 | 4.516 | -2.025 | -3.781 | -1.744 | 7.246 | -29.543 |
| **MM-GBSA of OMP21** | | | | | | | | |
| **0 ns** | -64.551 | -34.526 | 5.591 | -3.385 | -17.805 | -1.628 | 39.951 | -52.750 |
| **50 ns** | -46.957 | -25.684 | 2.397 | -1.716 | -13.071 | -0.653 | 30.876 | -39.105 |
| **100 ns** | -12.610 | -1.418 | 5.424 | -1.474 | -10.492 | -1.002 | 32.474 | -36.122 |

## 4 Conclusion

To summarize, this study unveils the discovery of new compounds through structural modifications, resulting in a wide range of alterations in physicochemical, biological, and pharmacokinetic properties through the incorporation of various functional groups. Certain alterations have been found to affect the dipole moment and free energy of many analogues. Thermodynamic studies evaluate the chemical, dielectric, and chemical durability, whereas UV-Vis spectral analysis visually depicts electronic transition spectra and absorption of the analogues at different wavelengths. In addition, FT-IR spectrum analysis offers information on the vibrational modes of analogues and, most significantly, identifies the presence of various functional groups in the molecular structure of the analogues. Nonbonding interactions, such as hydrogen and hydrophobic bonds, make certain analogues an effective therapeutic candidate. The ideal bond distances seen in many analogues are further strengthened by strong hydrogen bonding, raising affinity and highlighting the significance of nonbonding interactions in drug stability. After undertaking ADMET and PASS predictions on all analogues, it was ultimately determined that certain OMP analogues outperformed the control. Furthermore, the results of molecular docking, interactions, MD simulation, and principal component analysis make it evident that a few analogues, such as OMP3 and OMP21, exhibit better binding affinities to the targeted protein PTAAC1 than OMP. These in silico analyses serve as valuable tools in preclinical research, providing insights into advancing the development of a modified version of the primary studied drug with improved properties. However, experimental validation is necessary to verify potency, toxicity, off-target effects, and other pharmacokinetic and pharmacodynamic properties of these novel OMP analogues. Prospectively, validating these preclinical profiles will necessitate undertaking *in vivo* and *in vitro* experiments to bridge computational predictions with therapeutic significance.

## Supporting information

**S1 File.** **S1 Table.** Chemical structure and IUPAC name of the OMP and its analogues. **S1 Fig.** Optimized chemical structure of OMP and its analogues. **S2 Fig.** Molecular orbitals (HOMO and LUMO) and HUMO-LOMO energy gap of OMP and its analogues. **S3 Fig.** Free energies (Hartree) (a), dipole moments (Debye) (b), HOMO-LUMO gaps (c), and

chemical softness (d) of OMP and its analogues. **S2 Table.** Energy (eV) of HOMO-LUMO, gap, hardness (η), softness (S), chemical potential (µ), electronegativity (χ), and electrophilicity (ω) of OMP, and its analogues. **S4 Fig.** Electrostatic potential map of OMP analogues. **S5 Fig.** FT-IR spectra of OMP analogues. **S6 Fig.** UV-vis spectra of OMP analogues. **S3 Table.** Vibrational frequencies of OMP analogues. **S4 Table.** UV-vis spectral data of OMP and its analogues. **S7 Fig.** Binding energy of OMP and its analogues with the targeted PTAAC1 protein. **S5 Table.** Binding affinity and nonbonding interactionss of remaining compounds with the PTAAC1 protein. **S8 Fig.** Superimposed view of the docked conformer with the targeted protein, non-bonding interactions, and the hydrogen bond surface of OMP and its analogues. **S6 Table.** Absorption, distribution, metabolism, and toxicological properties (ADMET) studies of OMP analogues. **S7 Table.** PASS predicted data of OMP analogues.
(DOCX)

**S2 File. Protein RMSD.**
(XLSX)

**S3 File. Protein RMSF.**
(XLSX)

**S4 File. Ligand properties and H bonds.**
(XLSX)

**S5 File. Post-simulation MM-GBSA.**
(XLSX)

## Acknowledgement

The authors are thankful to 'Computer in Chemistry and Medicine Laboratory', Dhaka, Bangladesh, and 'Bioinformatics Laboratory (BioLab), Noakhali, Bangladesh', for their insightful guidelines and suggestions.

## Author contributions

**Conceptualization:** Monir Uzzaman.

**Data curation:** Mahmudul Hasan, Md. Ifteker Hossain, Noimul Hasan Siddiquee, Ezaz Ahmed, Md Rahamatolla, Tasrin Nahar, Popy Rani Paul, Mahmudul Hassan Suhag.

**Formal analysis:** Mahmudul Hasan, Md. Ifteker Hossain, Noimul Hasan Siddiquee, Ezaz Ahmed, Md Walid Hossain Talukder.

**Investigation:** Mahmudul Hassan Suhag, Monir Uzzaman.

**Methodology:** Mahmudul Hasan, Md. Ifteker Hossain, Noimul Hasan Siddiquee, Ezaz Ahmed, Md Walid Hossain Talukder, Md Rahamatolla, Tasrin Nahar, Popy Rani Paul, Monir Uzzaman.

**Project administration:** Monir Uzzaman.

**Resources:** Mahmudul Hasan, Md. Ifteker Hossain, Noimul Hasan Siddiquee.

**Software:** Mahmudul Hasan, Md. Ifteker Hossain, Noimul Hasan Siddiquee.

**Supervision:** Monir Uzzaman.

**Validation:** Mahmudul Hasan, Md. Ifteker Hossain, Noimul Hasan Siddiquee, Ezaz Ahmed, Monir Uzzaman.

**Visualization:** Mahmudul Hasan, Md. Ifteker Hossain, Noimul Hasan Siddiquee, Ezaz Ahmed, Md Walid Hossain Talukder.

**Writing – original draft:** Mahmudul Hasan, Md. Ifteker Hossain, Noimul Hasan Siddiquee, Ezaz Ahmed, Mahmudul Hassan Suhag.

**Writing – review & editing:** Mahmudul Hasan, Md. Ifteker Hossain, Noimul Hasan Siddiquee, Ezaz Ahmed, Md Rahamatolla, Tasrin Nahar, Popy Rani Paul, Mahmudul Hassan Suhag, Monir Uzzaman.

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
