## [Decision Letter · Decision Letter 0]

19 Mar 2025

PONE-D-25-11516Design of omeprazole derivatives for enhanced proton pump inhibition of potassium-transporting ATPase alpha chain 1; a spectrochemical, medicinal and pharmacological studyPLOS ONE

Dear Dr. Uzzaman,

Thank you for submitting your manuscript to PLOS ONE. After careful consideration, we feel that it has merit but does not fully meet PLOS ONE’s publication criteria as it currently stands. Therefore, we invite you to submit a revised version of the manuscript that addresses the points raised during the review process.

**ACADEMIC EDITOR: **The revised version of the article “Design of omeprazole derivatives for enhanced proton pump inhibition of potassium-transporting ATPase alpha chain 1; a spectrochemical, medicinal and pharmacological study” was assessed for meeting the standards for publication in PLOS ONE. The authors have demonstrated promising inhibitory potential of some structurally designed omeprazole (OMP1-OMP22) on the alpha chain 1 of potassium-transporting ATPase. However, the scope of the study is per below the title. For instance, the authors have mentioned “spectrochemical study” in the title whereas no content of the manuscript could be identified as spectrochemical analysis or results. Similarly, the terms medicinal and pharmacological study appear superfluous in the absence of experimental analysis. More importantly, how did the authors conduct the structural modifications to obtain the OMP derivatives? By mere drawing? I wonder how the feasibility of existence and stability of the derivatives can be ascertained in the absence of real or simulated tests. Instead, the authors could have searched for substructures of OMP in the chemical databases, and screen for analysis. With these, the feasibility of existence and stability of the identified derivatives can be ascertained for further study. These grey areas and those highlighted by the reviewers need to be seriously clarified before the submission can be reconsidered. . 

We look forward to receiving your revised manuscript.

Kind regards,

Yusuf Oloruntoyin Ayipo, Ph.D

Academic Editor

PLOS ONE

Additional Editor Comments:

The revised version of the article “Design of omeprazole derivatives for enhanced proton pump inhibition of potassium-transporting ATPase alpha chain 1; a spectrochemical, medicinal and pharmacological study” was assessed for meeting the standards for publication in PLOS ONE. The authors have demonstrated promising inhibitory potential of some structurally designed omeprazole (OMP1-OMP22) on the alpha chain 1 of potassium-transporting ATPase. However, the scope of the study is per below the title. For instance, the authors have mentioned “spectrochemical study” in the title whereas no content of the manuscript could be identified as spectrochemical analysis or results. Similarly, the terms medicinal and pharmacological study appear superfluous in the absence of experimental analysis. More importantly, how did the authors conduct the structural modifications to obtain the OMP derivatives? By mere drawing? I wonder how the feasibility of existence and stability of the derivatives can be ascertained in the absence of real or simulated tests. Instead, the authors could have searched for substructures of OMP in the chemical databases, and screen for analysis. With these, the feasibility of existence and stability of the identified derivatives can be ascertained for further study. These grey areas and those highlighted by the reviewers need to be seriously clarified before the submission can be reconsidered.

Reviewers' comments:

Reviewer's Responses to Questions

**Comments to the Author**

1. Is the manuscript technically sound, and do the data support the conclusions?

Reviewer #1: Yes

Reviewer #2: Yes

Reviewer #3: Yes

Reviewer #4: Partly

2. Has the statistical analysis been performed appropriately and rigorously? 

Reviewer #1: Yes

Reviewer #2: N/A

Reviewer #3: N/A

Reviewer #4: N/A

3. Have the authors made all data underlying the findings in their manuscript fully available?

Reviewer #1: Yes

Reviewer #2: Yes

Reviewer #3: Yes

Reviewer #4: Yes

4. Is the manuscript presented in an intelligible fashion and written in standard English?

Reviewer #1: Yes

Reviewer #2: Yes

Reviewer #3: Yes

Reviewer #4: Yes

5. Review Comments to the Author

Reviewer #1: Strengths:

- This study does a great job of providing a thorough computational analysis of omeprazole derivatives, successfully identifying three potential lead compounds: OMP3, OMP19, and OMP21.

- By using advanced computational techniques like DFT, TD-DFT, and MD simulations, the findings gain a higher level of reliability.

- The ADMET and PASS predictions offer important insights into the pharmacokinetics and safety profiles of the proposed compounds.

Areas for Improvement:

1. Experimental Validation: While the computational work is impressive, it's important to note that in vitro and in vivo tests are necessary to truly confirm the predicted efficacy and safety of these new derivatives. The authors should clearly mention this limitation.

2. Clarity and Readability: Some parts of the paper, especially the sections on computational methods, use complex terminology that could be simplified for better understanding. A few minor adjustments in language would enhance readability for a broader audience.

3. Discussion of Clinical Relevance: The manuscript would be even stronger if it included a comparison of how these derivatives relate to current PPIs in terms of clinical use, potential advantages, and associated risks.

4. Grammar and Formatting: There are a few minor grammatical issues that should be addressed, particularly in the introduction and discussion sections.

Final Recommendation:

- Some minor revisions are needed to improve clarity and emphasize the importance of experimental validation.

- Overall, the manuscript is well done and ready for publication after addressing the suggested changes.

Reviewer #2: Although the manuscript makes commendable use of a computational approach, the reliance on an AlphaFold-predicted structure for the potassium-transporting ATPase alpha chain 1 (PTAAC1) would benefit from explicit mention of the model’s confidence metrics. Stating the predicted local distance difference test (pLDDT) score or any reference to existing crystallographic data would strengthen confidence in docking results.

The docking scores of certain omeprazole analogs are indeed superior to the original omeprazole. However, readers would welcome a brief comparison to known commercial PPIs such as pantoprazole or esomeprazole. Such a comparison would situate these findings in a broader clinical context and highlight the magnitude of improvement.

The study thoroughly reports parameters like RMSD, RMSF, and rGyr, but it does not discuss thresholds that would characterize significant differences. Offering a straightforward explanation of how these metrics demonstrate genuine stability, beyond standard random fluctuations, would clarify the practical impact of the observed numerical variations.

The analyses reveal that functional groups such as –OCH3, –OCF3, –NH2, and –NHCOCH3 influence the dipole moments, free energies, and HOMO-LUMO gaps. A concise discussion explaining how each group’s electron-withdrawing or -donating nature influences both reactivity and predicted pharmacological properties would strengthen the connection to medicinal chemistry principles.

The work relies on in silico methods, yet the text does not elaborate on the next steps in experimental verification. Explaining how these analogs might be further validated, whether through biochemical assays, cell culture studies, or animal models, would highlight translational relevance and guide future researchers toward practical implementation of the findings.

Reviewer #3: Thank you for the opportunity to peer-review this article titled “Design of omeprazole derivatives for enhanced proton pump inhibition of potassium-transporting ATPase alpha chain 1; a spectrochemical, medicinal and pharmacological study.” The article described three major omeprazole derivatives that were identified through molecular docking with potassium transporting enzymes to demonstrate improved pharmacological efficacy and improved side effect profiles over the conventional omeprazole. This will create an opportunity for future research into these agents for better alternatives to the current omeprazole.

Overview

The overall idea of this article was well explained and structured as contained in the abstract. The methods used were adequate to support their findings and give rooms for future opportunities to research around proton pump inhibitors (PPI)-medications with enhanced clinical benefits plus little or no side effects. Each section of the article such as abstract, introduction, methodology, results, and conclusion were adequately written and presented or documented in clear terms. Similarly, other sections like conflict of interest, funding, clinical trial number, data availability, consent for publication, ethical/consent for approval, acknowledgement, and other author’s competing interests were adequately addressed. However, the authors did not identify any limitation to this study and few other observations which shall be discussed under each section.

Introduction

This section was well captured by the authors. However, the authors should reword these words “health goods” in the opening sentence of the first paragraph under introduction to properly convey their intent and avoid ambiguity. Also, paragraph three is not consistent with the ‘reference number 18.’ Similarly, paragraphs three, four and five need to be properly cited. This statement in paragraph five needs to be restructured regarding the quoted reference “It is conceivable to develop a superior drug alternative by structurally altering chemical compounds (34–36).”

Methodology

The authors need to pay attention to spelling error “(2.1 Geometry Optimiation)”. The authors may need to look at line seven of the first paragraph under methodology for the quoted reference “(39-42)”, this was what the authors did; I don’t think the citation is necessary except the authors have justification for that. These citations should be addressed too: “These computations were performed based on the Parr and Pearson interpretation (43–46), and Koopmans' theorem (47)”. The authors should try and reference the equation to determine hardness, softness, and chemical potential as well as the molecular formular table.

Results and Discussion

The results were well documented with some literature backups. The authors should try and add the relevance of Radius of Gyration and Root Mean Square Fluctuation analysis to these three omeprazole derivatives of interest.

Conclusion

Good conclusion. The authors need to state if the observed properties of these omeprazole derivatives after subjecting them into those various analyses and modifications will make them to have better pharmacological and improved side effects profiles over the conventional medication as contained in the abstract section. Also, the authors may need to rewrite the statement in line fourteen “it was found that certain ones displayed superior results compared to others.”

Recommendations

I recommend that this article be accepted for publication after the relevant corrections must have been made.

Reviewer #4: Please make the font size in figures readable. I am unable to read the details in the figures.especially 9 and 11.Please explain the figures in legends.why 100 ns simulations were recorded?

How can we test OMP3 and OMP21 as potential for next generation PPIs and how can we say that after computational modeling these PPIs have no side effects or less side effects?

6. PLOS authors have the option to publish the peer review history of their article (what does this mean? ). If published, this will include your full peer review and any attached files.

**Do you want your identity to be public for this peer review?** For information about this choice, including consent withdrawal, please see our Privacy Policy .

Reviewer #1: No

Reviewer #2: No

Reviewer #3: **Yes: ** Idowu Peter Shileayo Adebayo

Reviewer #4: **Yes: ** Shelly Saima Yaqub

---

## [Author Response · Author response to Decision Letter 1]

1 May 2025

Response to the reviewers:

We would like to thank all the reviewers as well as editor for their insightful comments.

Additional Editor Comments to the Author:

The authors have mentioned “spectrochemical study” in the title whereas no content of the manuscript could be identified as spectrochemical analysis or results. Similarly, the terms medicinal and pharmacological study appear superfluous in the absence of experimental analysis. More importantly, how did the authors conduct the structural modifications to obtain the OMP derivatives? By mere drawing? I wonder how the feasibility of existence and stability of the derivatives can be ascertained in the absence of real or simulated tests. Instead, the authors could have searched for substructures of OMP in the chemical databases, and screen for analysis. With these, the feasibility of existence and stability of the identified derivatives can be ascertained for further study. These grey areas and those highlighted by the reviewers need to be seriously clarified before the submission can be reconsidered.

Response: We acknowledge the reviewer's concern about the title's reference to the spectrochemical study. Spectrochemical studies refer to the use of spectroscopic techniques such as FT-IR and UV-Vis spectroscopy to investigate a chemical substance's composition, structure, and properties. Our studies include FT-IR, UV-Vis spectroscopy, and analysis to examine the structural and electronic properties of the designed omeprazole derivatives. These techniques are essential in drug design to confirm the molecular structure, assess to help in reactivity and stability as well as electronic properties (e.g., HOMO-LUMO gap) and interaction with the targeted protein, and confirm the presence of modified functional groups in the side chain.

By mentioning the 'designing of OMP analogues,' we intended to signify the in silico aspect of our study, which is why we refrained from directly using the term 'computational pharmacological analysis.' Our ADMET analysis, an essential and commonly employed method in drug development, offers insights into the pharmacokinetic and toxicity characteristics of the synthesised derivatives, facilitating evaluations of drug-likeness, bioavailability, and safety.

The structural modifications of OMP derivatives were not arbitrarily drawn but designed using rational drug design principles. Functional groups were carefully inserted to improve omeprazole's pharmacokinetic and pharmacodynamic aspects while keeping its core structure. To assure feasibility and stability, we performed quantum chemistry and thermodynamic investigations to determine electronic characteristics and energetic favorability. FT-IR and UV-Vis spectroscopy confirmed functional groups and structural integrity, whereas ADMET and PASS predictions assessed drug likeness and toxicity. Molecular docking evaluated binding interactions, while MD simulations confirmed enzyme-ligand stability under physiological settings. While substructure searches in chemical databases are one technique, our computational assessments revealed that the proposed derivatives are chemically viable and stable, confirming their potential for further investigation.

Reviewer #1:

Areas for Improvement:

1. Experimental Validation: While the computational work is impressive, it's important to note that in vitro and in vivo tests are necessary to truly confirm the predicted efficacy and safety of these new derivatives. The authors should clearly mention this limitation.

Response: As no funding has been received for this research, our study was limited to in silico analyses, and we could not proceed with experimental validation through in vitro or in vivo studies. We added this limitation to the conclusion of the revised manuscript.

2. Clarity and Readability: Some parts of the paper, especially the sections on computational methods, use complex terminology that could be simplified for better understanding. A few minor adjustments in language would enhance readability for a broader audience.

Response: According to your suggestion, we simplified the complex terminologies in the revised manuscripts.

3. Discussion of Clinical Relevance: The manuscript would be even stronger if it included a comparison of how these derivatives relate to current PPIs in terms of clinical use, potential advantages, and associated risks.

Response: In this study, we used omeprazole (a first-generation PPI) as the control and maintained it consistently across all analyses. Following your comment, we conducted a literature review to find relevant comparative data with other marketed PPIs but didn't find information aligning directly with our study framework. To address this, we recalculated ADMET values, which showed nearly identical results for omeprazole and esomeprazole (BBB = –0.6326, HIA = +0.9968, Caco-2 = +0.8867) and slightly different values for pantoprazole (BBB = –0.5777, HIA = +0.9935, Caco-2 = +0.7262). These confirm the appropriateness of our selected control. Our primary aim was to assess how omeprazole-derived analogues perform relative to the parent compound. The thermochemical and spectrochemical studies revealed that several OMP analogues possess good reactivity, stability, ADMET and PASS values, and strong affinity to the targeted protein. Nonetheless, we recognize the value of broader comparisons and plan to include detailed evaluations with other marketed PPIs in future studies. I hope you will consider this.

4. Grammar and Formatting: There are a few minor grammatical issues that should be addressed, particularly in the introduction and discussion sections.

Response: We have addressed the suggested grammatical corrections in both the introduction and discussion sections. Additionally, we thoroughly reviewed and revised the entire manuscript to ensure overall grammatical accuracy.

Reviewer #2:

Although the manuscript makes commendable use of a computational approach, the reliance on an AlphaFold-predicted structure for the potassium-transporting ATPase alpha chain 1 (PTAAC1) would benefit from explicit mention of the model’s confidence metrics. Stating the predicted local distance difference test (pLDDT) score or any reference to existing crystallographic data would strengthen confidence in docking results.

Response: We have incorporated the pLDDT values in the manuscript, which were> 90, signifying a very high level of model confidence.

The docking scores of certain omeprazole analogs are indeed superior to the original omeprazole. However, readers would welcome a brief comparison to known commercial PPIs such as pantoprazole or esomeprazole. Such a comparison would situate these findings in a broader clinical context and highlight the magnitude of improvement.

Response: In response to your suggestion, we initially searched the literature for relevant comparative data on the docking scores of omeprazole analogues with commercial PPIs like pantoprazole and esomeprazole. Unfortunately, we could not find direct comparative studies or docking scores for these compounds in the available literature. As a result, we recalculated the docking scores for pantoprazole and esomeprazole alongside the omeprazole analogues and mentioned them in the revised manuscript. The calculated binding energies for pantoprazole, esomeprazole, and omeprazole were -7.2 kcal/mol, -7.3 kcal/mol, and -7.0 kcal/mol, respectively, in association with the targeted PTAAC1 protein. Our results revealed that certain omeprazole analogues exhibit superior docking scores compared to omeprazole and the commercially available PPIs pantoprazole and esomeprazole, suggesting potential improvements in binding affinity and therapeutic potential.

The study thoroughly reports parameters like RMSD, RMSF, and rGyr, but it does not discuss thresholds that would characterize significant differences. Offering a straightforward explanation of how these metrics demonstrate genuine stability, beyond standard random fluctuations, would clarify the practical impact of the observed numerical variations.

Response: Lower RMSD, RMSF, and rGyr values are typically associated with greater stability of the molecular complex. In our analysis, the values for these parameters were consistent with those of the control, indicating that the omeprazole analogues exhibit similar structural stability. This suggests their potential as effective alternatives to conventional PPIs. To clarify the relevance of these findings, we have verified that the observed variations are within acceptable thresholds, ensuring that the differences reflect true stability rather than random fluctuations. Furthermore, we have addressed the queries raised in individual sections, explaining how these metrics demonstrate genuine stability and their implications in the context of our study.

The analyses reveal that functional groups such as –OCH3, –OCF3, –NH2, and –NHCOCH3 influence the dipole moments, free energies, and HOMO-LUMO gaps. A concise discussion explaining how each group’s electron-withdrawing or -donating nature influences both reactivity and predicted pharmacological properties would strengthen the connection to medicinal chemistry principles.

Response: We have thoroughly reviewed the relevant literature to better understand and explain how specific functional groups such as –OCH₃, –OCF₃, –NH₂, and –NHCOCH₃—influence key electronic parameters including dipole moments, free energies, and HOMO-LUMO energy gaps. In the revised manuscript, we have incorporated a more detailed discussion highlighting how the electron-donating or -withdrawing nature of these substituents affects the compounds’ chemical reactivity and predicted pharmacological behavior. The electron-donating or withdrawing effect of functional groups has an impact on the dipole moment, free energy, and HOMO-LUMO gaps. The strong electron-withdrawing nature of fluorine leads to an increase in dipole moment in the presence of electronegative atoms or polar bonds. The trifluoromethoxy (–OCF3) group exhibits the largest dipole moment, followed by acetamido group (–NHCOCH3), which contains both a polar carbonyl and an amide group. Through the donation of their lone pairs and the influence of resonance, the methoxy (–OCH3) and amino (–NH2) groups contribute to an increase in the dipole moment. Regarding free energy, electron-donating groups (–OCH3, –NH2) contribute to the stabilization of the molecule, leading to a decrease in free energy; in contrast, electron-withdrawing groups (–OCF3, –NHCOCH2) result in the destabilization of the molecule, thereby increasing free energy. About the HOMO-LUMO gap, the presence of electron-donating groups pushes electro into the system, therefore raising the HOMO level, decreasing the gap, and increasing molecular reactivity, while electron-withdrawing groups increase the gap by pulling electrons from the system, stabilizing the molecule and reducing its reactivity. While –OCF3 and –NHCOCH3 lead to an expansion of the HOMO-LUMO gap, thereby enhancing stability, –OCH3 and –NH2 contribute to an overall increase in reactivity. These medicinal chemistry-based modifications maximize pharmacological effects and reactivity for stability and drug efficacy. We clarified these problems in the revised work to strengthen links between medication design and other aspects.

The work relies on in silico methods, yet the text does not elaborate on the next steps in experimental verification. Explaining how these analogs might be further validated, whether through biochemical assays, cell culture studies, or animal models, would highlight translational relevance and guide future researchers toward practical implementation of the findings.

Response: We appreciate your suggestion to elaborate on the next steps for experimental validation of our findings. However, the lack of financing for this work is a significant challenge for us in carrying out experimental validations. We would like to request you to consider this limitation.

Reviewer #3:

Overview

The overall idea of this article was well explained and structured as contained in the abstract. The methods used were adequate to support their findings and give rooms for future opportunities to research around proton pump inhibitors (PPI)-medications with enhanced clinical benefits plus little or no side effects. Each section of the article such as abstract, introduction, methodology, results, and conclusion were adequately written and presented or documented in clear terms. Similarly, other sections like conflict of interest, funding, clinical trial number, data availability, consent for publication, ethical/consent for approval, acknowledgement, and other author’s competing interests were adequately addressed. However, the authors did not identify any limitation to this study and few other observations which shall be discussed under each section.

Response: We have addressed limitations in the conclusion part of the revised version.

Introduction

This section was well captured by the authors. However, the authors should reword these words “health goods” in the opening sentence of the first paragraph under introduction to properly convey their intent and avoid ambiguity. Also, paragraph three is not consistent with the ‘reference number 18.’ Similarly, paragraphs three, four and five need to be properly cited. This statement in paragraph five needs to be restructured regarding the quoted reference “It is conceivable to develop a superior drug alternative by structurally altering chemical compounds (34–36).”

Response: All recommendations were carefully reviewed and incorporated into the revised manuscript to ensure clarity and improve the quality of work.

Methodology

The authors need to pay attention to spelling error “(2.1 Geometry Optimiation)”. The authors may need to look at line seven of the first paragraph under methodology for the quoted reference “(39-42)”, this was what the authors did; I don’t think the citation is necessary except the authors have justification for that. These citations should be addressed too: “These computations were performed based on the Parr and Pearson interpretation (43–46), and Koopmans' theorem (47)”. The authors should try and reference the equation to determine hardness, softness, and chemical potential as well as the molecular formular table.

Response: We’ve thoughtfully revised the manuscript based on your valuable suggestions.

Results and Discussion

The results were well documented with some literature backups. The authors should try and add the relevance of Radius of Gyration and Root Mean Square Fluctuation analysis to these three omeprazole derivatives of interest.

Response: We incorporated the significance of the Radius of Gyration and Root Mean Square Fluctuation analysis in the revised manuscript.

Conclusion

Good conclusion. The authors need to state if the observed properties of these omeprazole derivatives after subjecting them into those various analyses and modifications will make them to have better pharmacological and improved side effects profiles over the conventional medication as contained in the abstract section. Also, the authors may need to rewrite the statement in line fourteen “it was found that certain ones displayed superior results compared to others.”

Response: In the conclusion section, we have already addressed the pharmacological benefits and improved effects over conventional medication. Additionally, we have revised the line as per your remarks.

Reviewer #4:

Please make the font size in figures readable. I am unable to read the details in the figures especially 9 and 11.Please explain the figures in legends. Why 100 ns simulations were recorded?

Response: We regenerated Figures 9 and 10 with high resolution in the result section and explained the figures in legends.

Based on evidence from the literature, 100 ns molecular dynamics (MD) simulations are commonly used and generally considered reliable enough to provide meaningful insights into molecular interactions and stability. However, running longer simulations would undoubtedly provide more robust insights. Due to limited access to high-resolution computational facilities, we conducted simulations for 100 only. We sincerely hope you’ll take

---

## [Decision Letter · Decision Letter 1]

15 May 2025

PONE-D-25-11516R1Design of omeprazole derivatives for enhanced proton pump inhibition of potassium-transporting ATPase alpha chain 1; a spectrochemical, medicinal, and pharmacological studyPLOS ONE

Dear Dr. Uzzaman,

Thank you for submitting your manuscript to PLOS ONE. After careful consideration, we feel that it has merit but does not fully meet PLOS ONE’s publication criteria as it currently stands. Therefore, we invite you to submit a revised version of the manuscript that addresses the points raised during the review process.

**ACADEMIC EDITOR: **Authors are recommended to resubmit the revised manuscript for reconsideration following a minor revision as rightly recommended by a reviewer.. 

We look forward to receiving your revised manuscript.

Kind regards,

Yusuf Oloruntoyin Ayipo, Ph.D

Academic Editor

PLOS ONE

Journal Requirements:

Additional Editor Comments:

Authors are recommended to resubmit the revised manuscript for reconsideration following a minor revision as rightly recommended by a reviewer.

Reviewers' comments:

Reviewer's Responses to Questions

**Comments to the Author**

1. If the authors have adequately addressed your comments raised in a previous round of review and you feel that this manuscript is now acceptable for publication, you may indicate that here to bypass the “Comments to the Author” section, enter your conflict of interest statement in the “Confidential to Editor” section, and submit your "Accept" recommendation.

Reviewer #5: All comments have been addressed

Reviewer #6: (No Response)

2. Is the manuscript technically sound, and do the data support the conclusions?

Reviewer #5: Yes

Reviewer #6: Yes

3. Has the statistical analysis been performed appropriately and rigorously? 

Reviewer #5: Yes

Reviewer #6: Yes

4. Have the authors made all data underlying the findings in their manuscript fully available?

Reviewer #5: Yes

Reviewer #6: Yes

5. Is the manuscript presented in an intelligible fashion and written in standard English?

Reviewer #5: Yes

Reviewer #6: Yes

6. Review Comments to the Author

Reviewer #5: 1. The abstract lacks the study's objective and hypothesis; authors should incorporate this crucial information.

2. While the authors provided a rationale for their choice of manuscript title, terms like 'medicinal and spectroscopic' could be misleading here, as this manuscript doesn't include any wet lab studies. It's recommended that authors use terms like "computational" or "in silico" instead.

3. The authors did an excellent job evaluating the stability of the new analogues through various computational assessments. However, do these new analogues retain the potency and efficacy of the OMP? What about toxicity and off-target effects? Authors should address these questions, and if they can't, they should mention this in the limitations or future directions. Please keep in mind that a high docking score does not directly predict biological activity.

4. Several references are outdated and should be revised, particularly those cited from the 1980s and 1990s.

Reviewer #6: I agree that modifying the molecular structure of existing drugs is a strategic approach to enhancing their pharmacological properties, potentially leading to more effective and safer therapeutics. Structural alterations can influence a drug’s electronic distribution, polarity, hydrogen bonding potential, and steric characteristics, which in turn affect its binding affinity, solubility, metabolic stability, and bioavailability. By rationally designing drug candidates with optimized molecular composition and substituent positioning, researchers can improve target specificity, minimize adverse effects, and enhance therapeutic efficacy. This study has effectively achieved its aim to develop novel therapeutic alternatives by refining the structural features of OMP to improve its potency, efficacy, and safety profile.

I however, suggest that the title of this manuscript be edited. While this study has adequately used spectrochemical, as well as medical chemistry and pharmacological analysis in the structural modification and refining of OMP, all of these analysis were done computationally therefore, I recommend that the title reflect that the analysis were done computationally.

Also, are any of the newly designed OMP analogues novel? This should be included in the manuscript.

As regards structural modification and optimization, I understand that the core structure of OMP was maintained, and structural modification was carried out only on R1,R2 and R3. There are three additional positions on the benzene ring of the benzimidazole moiety and one additional position on the pyridine moiety open for functional group modifications why didn't you explore any of those positions? This question would be answered using the docking studies and analysis of the protein binding site.

Other than the above recommendations, this manuscript is in good shape and congratulations for excellent research and well written manuscript.

7. PLOS authors have the option to publish the peer review history of their article (what does this mean? ). If published, this will include your full peer review and any attached files.

**Do you want your identity to be public for this peer review?** For information about this choice, including consent withdrawal, please see our Privacy Policy .

Reviewer #5: No

Reviewer #6: No

---

## [Author Response · Author response to Decision Letter 2]

23 May 2025

Response To Reviewer

Reviewer #5:

1. The abstract lacks the study's objective and hypothesis; authors should incorporate this crucial information.

Response: Thank you for your response. We have incorporated the objectives and hypotheses into the abstract section of the current version of the manuscript. Also, considering the word limit, we had to summarize the objectives and hypothesis in one line and rewrite the abstract.

2. While the authors provided a rationale for their choice of manuscript title, terms like 'medicinal and spectroscopic' could be misleading here, as this manuscript doesn't include any wet lab studies. It's recommended that authors use terms like "computational" or "in silico" instead.

Response: Following your recommendation, we changed the title to “Computational design and cheminformatics profiling of omeprazole derivatives for enhanced proton pump inhibition of potassium-transporting ATPase alpha chain 1”

3. The authors did an excellent job evaluating the stability of the new analogues through various computational assessments. However, do these new analogues retain the potency and efficacy of the OMP? What about toxicity and off-target effects? Authors should address these questions, and if they can't, they should mention this in the limitations or future directions. Please keep in mind that a high docking score does not directly predict biological activity.

Response: In our in silico approach, we have evaluated and compared all the thermodynamic, spectrochemical and docking simulation results with the control, OMP, which is a marketed drug. To assess toxicity and off-target effects, we utilized the in silico ADMET and PASS predictions and compared them with omeprazole. We fully acknowledge the concern of the reviewer that in vivo and in vitro validation is necessary to portray the full picture of this preclinical research, which is our limitation and should be the future prospect. We have clearly mentioned this in the revised version of the conclusion, as per your recommendation.

4. Several references are outdated and should be revised, particularly those cited from the 1980s and 1990s.

Response: We agree that citing recent literature is vital, but these specific references (Refs. 43, 48, 53) represent the foundational works important to the methodological context. Subsequent research builds on but does not invalidate these foundational works. For reference 63, we undertook a comprehensive search but found no alternatives that validated our findings. We hope this clarification addresses your concerns, and appreciate your suggestions and recommendations.

Reviewer #6:

I agree that modifying the molecular structure of existing drugs is a strategic approach to enhancing their pharmacological properties, potentially leading to more effective and safer therapeutics. Structural alterations can influence a drug’s electronic distribution, polarity, hydrogen bonding potential, and steric characteristics, which in turn affect its binding affinity, solubility, metabolic stability, and bioavailability. By rationally designing drug candidates with optimized molecular composition and substituent positioning, researchers can improve target specificity, minimize adverse effects, and enhance therapeutic efficacy. This study has effectively achieved its aim to develop novel therapeutic alternatives by refining the structural features of OMP to improve its potency, efficacy, and safety profile. I however, suggest that the title of this manuscript be edited. While this study has adequately used spectrochemical, as well as medical chemistry and pharmacological analysis in the structural modification and refining of OMP, all of these analysis were done computationally therefore, I recommend that the title reflect that the analysis were done computationally.

Response: We deeply appreciate your constructive suggestion to refine the manuscript title for greater clarity. In accordance with your recommendation, we have revised the title to explicitly reflect the computational approaches conducted in this study. The updated title is now: “Computational design and cheminformatics profiling of omeprazole derivatives for enhanced proton pump inhibition of potassium-transporting ATPase alpha chain 1”

Also, are any of the newly designed OMP analogues novel? This should be included in the manuscript. As regards structural modification and optimization, I understand that the core structure of OMP was maintained, and structural modification was carried out only on R1, R2 and R3. There are three additional positions on the benzene ring of the benzimidazole moiety and one additional position on the pyridine moiety open for functional group modifications why didn't you explore any of those positions? This question would be answered using the docking studies and analysis of the protein binding site.

Response: To the best of our knowledge and literature research, none of the OMP analogues here have ever been studied, mentioned, or investigated before; hence, they are novel. As per your suggestion, we recognized the significance of describing them as novel and thus explicitly mentioned these analogues as 'novel' in the current revised version of the manuscript.

We appreciate the reviewer's insightful observation of our manuscript's unexplored positions on benzimidazole and pyridine moieties. The mentioned sites were also structurally modified and evaluated using thermodynamic, spectrochemical, and docking studies. However, these modifications showed no significant improvement in overall results, prompting a focus on candidates, which we have mentioned. Your consideration of our answer, rationale is much appreciated.

Other than the above recommendations, this manuscript is in good shape and congratulations for excellent research and well written manuscript.

Response: Thank you sincerely for your thoughtful feedback and generous appraisal of our work. We also appreciate your constructive suggestions to strengthen the alignment between the title, methodology, and overall refining the study.

---

## [Decision Letter · Decision Letter 2]

4 June 2025

Computational design and cheminformatics profiling of omeprazole derivatives for enhanced proton pump inhibition of potassium-transporting ATPase alpha chain 1

PONE-D-25-11516R2

Dear Dr. Uzzaman,

We’re pleased to inform you that your manuscript has been judged scientifically suitable for publication and will be formally accepted for publication once it meets all outstanding technical requirements.

Kind regards,

Yusuf Oloruntoyin Ayipo, Ph.D

Academic Editor

PLOS ONE

Additional Editor Comments (optional):

The submission meets the level of scientific rigour required for publication in this title and all the concerns raised by the respective reviewers have been addressed satisfactorily. I hereby recommend the current revised manuscript for publication.

Reviewers' comments:

Reviewer's Responses to Questions

**Comments to the Author**

1. If the authors have adequately addressed your comments raised in a previous round of review and you feel that this manuscript is now acceptable for publication, you may indicate that here to bypass the “Comments to the Author” section, enter your conflict of interest statement in the “Confidential to Editor” section, and submit your "Accept" recommendation.

Reviewer #6: All comments have been addressed

2. Is the manuscript technically sound, and do the data support the conclusions?

Reviewer #6: Yes

3. Has the statistical analysis been performed appropriately and rigorously? 

Reviewer #6: Yes

4. Have the authors made all data underlying the findings in their manuscript fully available?

Reviewer #6: Yes

5. Is the manuscript presented in an intelligible fashion and written in standard English?

Reviewer #6: Yes

6. Review Comments to the Author

Reviewer #6: (No Response)

7. PLOS authors have the option to publish the peer review history of their article (what does this mean? ). If published, this will include your full peer review and any attached files.

**Do you want your identity to be public for this peer review?** For information about this choice, including consent withdrawal, please see our Privacy Policy .

Reviewer #6: No

---

## [Editor Report · Acceptance letter]

PONE-D-25-11516R2

PLOS ONE

Dear Dr. Uzzaman,

I'm pleased to inform you that your manuscript has been deemed suitable for publication in PLOS ONE. Congratulations! Your manuscript is now being handed over to our production team.

Kind regards,

on behalf of

Dr. Yusuf Oloruntoyin Ayipo

Academic Editor

PLOS ONE